# EQUIVARIANT LATENT ALIGNMENT VIA FLOW MATCHING UNDER GROUP SYMMETRIES

## ABSTRACT

Geometry-aware generative models and novel view synthesis approaches have shown strong potential to improve visual fidelity and consistency. In parallel, equivariant representation learning has emerged as a powerful framework for constructing latent spaces where analytically known group transformations could act directly, capturing geometric structure in data and enhancing both interpretability and generalization in novel view synthesis. However, we identify that existing approaches often suffer from *latent misalignment*, a discrepancy between the intended group action and the actual required transformations in latent space, as the learned latents fail to consistently preserve the equivariant relations imposed by the underlying group symmetry. This misalignment degrades view synthesis quality and undermines the theoretical guarantees of equivariant representation learning. To address this issue, we introduce **Residual Latent Flow**, a flow-matching-based correction framework that corrects the misaligned latents, thereby improving compliance with the underlying equivariance relation. We show experiments that our method significantly reduces latent misalignment and improves novel view synthesis quality, under rotational groups $SO(n)$. Our method demonstrates the efficacy of combining flow-based correction with equivariant representation learning, resulting in learning a more consistent and accurate group symmetry-aware models.

## 1 INTRODUCTION

Recent advances in geometry-aware generative models, such as diffusion-based methods (Karnewar et al., 2023; Yu et al., 2023; Shi et al., 2023; Anciukevičius et al., 2023) and Generative Adversarial Networks (Chan et al., 2022), have significantly enhanced visual fidelity in generation. Moreover, geometry-aware novel view synthesis (NVS) approaches (Miyato et al., 2022; Koyama et al., 2023; Miyato et al., 2023) generate realistic images of scenes or objects from previously unseen viewpoints by leveraging geometric structure of its latent space, enabling consistent view synthesis and diverse computer vision applications.

In parallel, equivariant representation learning (Cohen & Welling, 2016; Falorsi et al., 2018; Dupont et al., 2020; Quessard et al., 2020) leverages intrinsic symmetries in dataset to enforce structured transformation behavior in the learned representations. By ensuring the latent representations to transform predictably under group actions (*e.g.*, rotation), such models offer improved generalization and interpretability of the latent space. This provides interpretable latent transformations aligned with the structure of compact Lie groups (Finzi et al., 2020; Ruhe et al., 2023).

Formally, a mapping $\Phi$ is *equivariant* if the mapping's representations *corotate* for a group action such as rotation is applied to the data: $\rho(g)\Phi(x) = \Phi(g \circ x)$, where $\rho$ is a pre-defined group representation for the elements of the interested symmetry group $g \in G$. Intuitively, the encoder $\Phi$ is equivariant if the latent upon the group action $\rho(g)\Phi(x)$ is aligned with the corresponding latent from the transformed image $\Phi(g \circ x)$.

An encoder-based equivariant representation learning relies on a strong assumption that the encoder $\Phi$ jointly learns both how to compress the object content and the underlying symmetry group's structure in a perfectly equivariant manner. In practice, a discrepancy between the analytically rotated latent $\rho(g)\Phi(x)$ and the true target latent $\Phi(g \circ x)$ could be induced by data variability

(*e.g.*, appearance, occlusion, or lighting). A more fundamental limitation also arises from the aliasing of intermediate feature representations. Prior works have shown that standard convolutional and transformer-based architectures inherently introduce aliasing due to discrete sampling and non-bandlimited filters, which leads to persistent equivariance errors even under idealized datasets (Karras et al., 2021; Azulay & Weiss, 2019; Rahaman et al., 2019). We succinctly refer to this issue as *latent misalignment*, which undermines the equivariance and degrades the fidelity of synthesized views.

To address this issue, we propose a principled latent correction mechanism **Residual Latent Flow**, which utilizes a learned flow as a bridge between analytic latents and true latents that satisfy equivariance relation. Although flow-based model has been mostly applied to transport Gaussian noise to the data distribution in generative models, its flexibility (Sec. 3.2) allows to transport between arbitrary distributions (Cohen et al., 2025; Fischer et al., 2025). Our main idea is to learn a transport from the analytic latent $\rho(g)\Phi(x)$ to its empirically encoded target $\Phi(g \circ x)$ using Flow Matching (Lipman et al., 2022; Liu et al., 2022; Albergo et al., 2023). Rather than discarding the group-theoretic prior, such as the well-established theory of Wigner $D$-matrix representation of the rotation group $SO(3)$ (Wigner, 2012; Rose, 1995; Sakurai & Napolitano, 2020), we treat the analytically known group representation $\rho(g)$ as a first-order approximation and employ flow matching to learn the residual terms, to realize transportation of the latents.

The main contributions can be summarized as follows:

- **Identification of latent misalignment in equivariant models.** We formally define and analyze the phenomena of *latent misalignment* that arise from discrepancies between analytically transformed and empirically encoded representations, and show how it undermines geometric consistency and synthesis fidelity in Sec. 3.1.

- **Latent representation correction via flow model**. We propose a novel latent correction method based on flow matching to align analytically rotated latents with their encoded targets, enabling smooth and data-driven correction, while preserving group structure in Sec. 3.2.

- **Improvements in consistency and synthesis quality.** We demonstrate that our method improves latent alignment and enhances image reconstruction accuracy across multiple datasets that inherently admit rotational symmetries, such as $SO(2)$ and $SO(3)$, in Sec. 4.3.

## 2 BACKGROUND

### 2.1 EQUIVARIANT REPRESENTATION LEARNING UNDER GROUP SYMMETRY

Equivariant representation learning (ERL) leverages symmetry properties inherent in data to capture latent structural relationships. Formally, a mapping $\Phi : x \in X \mapsto \Phi(x)$ is equivariant with respect to a group $G$ if it satisfies the following relation:

$$\Phi(g \circ x) = \rho(g)\Phi(x), \quad \forall x \in X, \quad \forall g \in G, \quad (1)$$

where $\circ : G \times X \to X$ is the group action on the set of data $X$, $\rho : G \to GL(n, \mathbb{R})$ is the group representation of group $G$. Fig. 1 pictorially explains the equivariance relation. Then, a complementary decoder $\Psi$ targets to recover the original input from the latent representation: $\Psi(\Phi(x)) = x, \forall x \in X$.

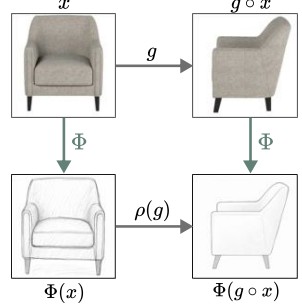

Figure 1: Illustration of equivariance relation. Mapping $\Phi$ is equivariant iff $\Phi(g \circ x) = \rho(g)\Phi(x)$.

In practice, ERL aims to ensure that latent features transform consistently with the underlying group action. To achieve this, $\Phi$ and $\Psi$ are trained with the equivariance loss, which explicitly penalizes deviations from the equivariance relation, and a reconstruction loss, which encourages preservation of sufficient information to faithfully recover the input. Formally,

$$\mathcal{L}_{\mathrm{ERL}} = \underbrace{\mathbb{E}[\|\Phi(g \circ x) - \rho(g)\Phi(x)\|_2^2]}_{\text{Equivariance Loss}} + \underbrace{\mathbb{E}[\|g \circ x - \Psi(\rho(g)\Phi(x))\|_2^2]}_{\text{Decoder Loss}}. \quad (2)$$

## 2.2 Special Orthogonal Groups

In this work, we focus on the datasets that explicitly include controlled group action, specifically rotation, *e.g.*, turntable scans or object-centric synthetic renders. These are collected for novel view synthesis (NVS) as multi-view captures of a scene, which can be naturally modeled by the special orthogonal groups SO(n).

**SO(3).** The spherical orthogonal group SO(3) is the group of rotations in three-dimensional space, defined as

$$\mathrm{SO}(3) = \{\mathbf{R} \in GL(3, \mathbb{R}) \mid \mathbf{R}^\top \mathbf{R} = \mathbf{R}\mathbf{R}^T = I, \ \det(\mathbf{R}) = 1\}. \tag{3}$$

The group elements can be parametrized with three Euler angles $\alpha, \beta, \gamma$, as $R(\alpha, \beta, \gamma) = e^{-i\alpha J_z} e^{-i\beta J_y} e^{-i\gamma J_z}$, where $J_x, J_y, J_z$ are the generators (angular momentum operators) of the Lie algebra $\mathfrak{so}(3) = \{\mathbf{A} \in GL(3, \mathbb{R}) \mid \mathbf{A}^\top = -\mathbf{A}\}$ of SO(3). In parallel, there exists Wigner $D$-matrix representation $D^{(\ell)} : \mathrm{SO}(3) \to GL(2\ell + 1, \mathbb{C})$ of degree $\ell$ that maps the rotation group element to a $(2\ell + 1) \times (2\ell + 1)$ matrix: $D_{m,n}^{(\ell)} = e^{-im\alpha} d_{m,n}^{(\ell)}(\beta) e^{-in\gamma}$, where $d_{m,n}^{(\ell)}(\beta)$ is the real-valued Wigner small-$d$ matrix depending on the polar angle $\beta$ and $m, n \in \{-\ell, \cdots, \ell\}$ index the basis states of the degree-$\ell$ irreducible representation. Refer to Sec. B for more details.

**SO(2).** The special orthogonal group SO(2),

$$\mathrm{SO}(2) = \{\mathbf{R} \in GL(2, \mathbb{R}) \mid \mathbf{R}^\top \mathbf{R} = I, \ \det(\mathbf{R}) = 1\}, \tag{4}$$

is a subgroup of SO(3) consisting of rotations about the $z$-axis. The irreducible unitary representations of SO(2) are all one-dimensional characters, indexed by an integer frequency $m \in \mathbb{Z}$: $\rho_m(\theta) = e^{im\theta}$. These characters arise naturally by restricting the Wigner $D$-matrices of SO(3) (*i.e.*, $\beta = \gamma = 0$ and $\alpha = \theta$). Refer to Sec. B for more details.

## 2.3 Flow Matching

Flow matching is a generative modeling framework that seeks to transform samples from a prior distribution $p_0$ to a target distribution $q$ through a continuous-time flow induced by an ordinary differential equation (ODE):

$$\frac{d}{dt}\psi_t(z) = v_t(\psi_t(z)), \tag{5}$$

where flow $\psi : (t, z) \mapsto \psi_t(z) = z_t$ is a time-dependent diffeomorphism that pushforwards $p_0$ to $q$ and velocity field $v : (t, z) \mapsto v_t(z)$ is a solution to the ODE. Then, given a density $p_0$ at $t = 0$, the probability path $p_t : \mathbb{R}^d \to \mathbb{R}$ can be identfied as the pushforward of $p_0$ under flow, $p_t = \psi_t^\# p_0$. Since the true marginal $v_t$ is intractable, Lipman et al. (2022); Liu et al. (2022); Albergo et al. (2023) introduce a simulation-free training framework using conditional flow matching loss:

$$\mathcal{L}_{\mathrm{CFM}}(\theta) = \mathbb{E}_{t \sim U[0,1], \, \varepsilon \sim q(\cdot), \, z_t \sim p_t(\cdot|c)} \left[ \|v_\theta(t, z_t) - v_t(z_t|\varepsilon)\|_2^2 \right], \tag{6}$$

# 3 Method

## 3.1 Motivation: Latent Misalignment in Equivariant Representation Learning

**Base ERL Framework.** We employ Neural Fourier Transform (NFT) (Koyama et al., 2023; Miyato et al., 2022) for our analysis and experiments. Specifically, NFT considers a basis transform $P$ that block diagonalizes the group representation $\tilde{\rho}$ of a given group $G$, facilitating the decomposition into $n$ irreducible components:

$$\rho(g) = \bigoplus_{i=1}^{n} \rho_i(g), \qquad \rho(g) = P\tilde{\rho}(g)P^{-1}, \tag{7}$$

where $\rho$ is the block-diagonal representation after applying similarity transformation to $\tilde{\rho}$. Then $\rho$ can be identified as a direct sum of irreducible representations $\rho_i$. The $\ell$-th irreducible block of the representation coincides with the Wigner $D$-matrix of degree-$\ell$, *i.e.*, $\rho_\ell(g) = D^{(\ell)}(g)$. The NFT framework provides interpretable block components via block-level equivariant learning, offering a fine-grained evaluation.

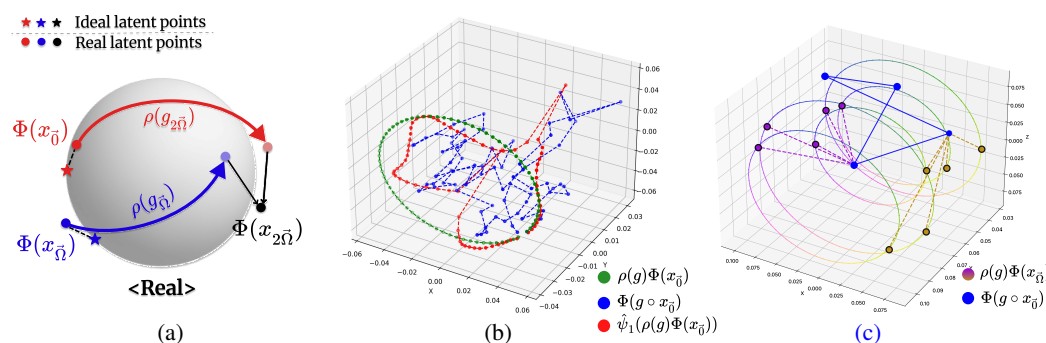

Figure 2: **(a) Illustration of empirical latent trajectories under group transformations.** In practice, real latent trajectories (circles) deviate from the ideal ones (stars), resulting in inconsistent endpoints. Two images obtained by viewing single object from two different angles are depicted as $x_{\vec{0}}$ and $x_{\vec{\Omega}}$. **(b) Visualization of latent trajectories under SO(3) degree-1 rotation block.** Green dots and blue dots depict analytically transformed latents $\rho(g)\Phi(x_{\vec{0}})$ and encoder-derived latents $\Phi(g \circ x_{\vec{0}})$, respectively. Red dots depict corrected latents by our method. **(c) Motivation to use flow matching.** Each cyclic trajectory (colored with a smooth cyclic colormap) corresponds to the latent orbit of different initial views, $\mathrm{Orb}(\Phi(x_{\vec{\Omega}_i})) := \{\rho(g)\Phi(x_{\vec{\Omega}_i}) \mid g \in G\}$. The blue target latents, $\Phi(g_j \circ x_{\vec{0}})$, and their corresponding source latents (other colors), $\rho(g_j g_{\vec{\Omega}_i}^{-1})\Phi(x_{\vec{\Omega}_i})$, are misaligned in practice. We can resolve this as a distribution transport problem, utilizing flow matching.

After representing the group elements $g \in G$ as a direct sum of degree-$\ell$ Wigner $D$-matrix, we are interested in training an autoencoder that learns to map data to a latent representation such that *corotate* for a group action: $\rho(g)\Phi(x) = \Phi(g \circ x)$. Hence, given the group representation $\rho : G \to \oplus_{\ell=0}^{L} GL(\dim(\rho_\ell), \mathbb{R})$, where $L$ is the maximum degree of the representations, we consider training an encoder $\Phi : \mathbb{R}^{H \times W} \to \mathbb{R}^{C \times N_G}$ and a decoder $\Psi : \mathbb{R}^{C \times N_G} \to \mathbb{R}^{H \times W}$ such that minimize the ERL loss (Eq. (2)). Here, $H, W$ denote the dimension of input image, $C$ is the latent representation's channel size, and $N_G$ is the sum of dimensions of all degree-$\ell$ representations of group $G$. For example, in SO(2), $N_{\mathrm{SO(2)}} = 1 + 2L$ because the group elements comprise one scalar block for trivial representation ($\ell = 0$) and $L$ $2 \times 2$ rotation blocks for other non-zero degree representation. In SO(3), $N_{\mathrm{SO(3)}} = \sum_{\ell=0}^{L}(2\ell + 1) = (L+1)^2$ which is a dimension of concatenation of all degree-$\ell$ vectors of SO(3).

**Latent Misalignment.** As illustrated in Fig. 2b, the actual latent paths (blue) learned by the encoder form irregular, jagged trajectories, far from the analytically derived paths (green). In other words, for the same object, analytically rotated $\rho(g_{\Delta\theta})\Phi(x_\theta)$ and true encoding $\Phi(x_{\theta+\Delta\theta})$ diverges. We further observe that this discrepancy grows with rotation magnitude $\Delta\theta$, implying accumulation of the error (See Fig. 4).

Correcting this latent misalignment is crucial, since it undermines the consistency and interpretability of the learned representations. The latent misalignment naturally arises because as the analytical transformation $\rho(g)$ is a fixed linear operator applied in the latent space, it lacks the expressiveness to model fine-grained visual effects such as self-occlusion, lighting variation, or subtle texture changes that arise from real 3D transformations. See Sec. I for more discussions about the cause of latent misalignment.

This highlights the need for a *correction mechanism* for refining the learned equivariant latent representations. To this end, we aim to mitigate this gap in reality by introducing a principled transportation mechanism while preserving the group-theoretic foundation.

### 3.2 Residual Latent Flow for Equivariant Representation Correction

Based on our observations of latent misalignments in ERL, we propose **Residual Latent Flow**, a method for latent representation correction based on flow matching (Lipman et al., 2022; Liu et al., 2022; Albergo et al., 2023). Specifically, we employ the analytically transformed latent $\rho(g)\Phi(x)$ as a first-order approximation and let continuous time flow to transport this latent to its corresponding target encoding $\Phi(g \circ x)$, while preserving the group structure.

Figure 3: **Illustration of our Residual Latent Flow.** Standard encoder-based equivariant representation learning frameworks suffer from *latent misalignment*, where the learned latent codes do not align with the intended equivariant structure, i.e. $\rho(g)\Phi(x) \neq \Phi(g \circ x)$ for transformation $g \in G$. Our method introduces a flow matching based latent correction step that explicitly realigns latents. This correction step explicitly enforces alignment $\rho(g)\Phi(x) \approx \Phi(g \circ x)$, which restores consistency in the latent space and improves visual fidelity in novel view synthesis.

**Problem Formulation.** Let $(x, g) \sim q$ be a data pair sampled from a dataset, where $x$ is an image and $g \in G$ is a known group transformation (*e.g.*, a rotation). Let us denote the latent representation of the original image by $\Phi(x)$. Then, the analytic group transformation applied to latent ($z_0$) and the latent of the transformed image ($z_1$) can be denoted as

$$z_0 := \rho(g)\Phi(x), \qquad z_1 := \Phi(g \circ x), \tag{8}$$

respectively. We refer to the distributions of $z_0$ and $z_1$ as the source distribution $p_0(z_0)$ and target distribution $p_1(z_1)$, respectively. As shown in Sec. 3.1, $z_0$ and $z_1$ are not precisely aligned in practice due to the imperfect encoder and data variability. To correct this misalignment, we connect the $z_0$ to its corresponding $z_1$ via learned flow $\psi : [0, 1] \times \mathbb{R}^d \to \mathbb{R}^d$.

**Latent Correction by Flow Matching.** To achieve this, we need to design the probability path which starts at $z_0$ and ends at $z_1$. Considering this boundary conditions, we derive the marginal probability path by marginalizing conditional probability paths:

$$p_t(z_t) = \iint p_t(z_t|z_0, z_1) \, \pi_{0,1}(z_0, z_1) \, dz_0 \, dz_1, \quad z_0 \sim p_0(z_0), \quad z_1 \sim p_1(z_1), \tag{9}$$

where $\pi_{0,1}$ is the joint distribution of the variables $z_0, z_1$. However, unlike the previous flow matching frameworks with arbitrary priors (Liu et al., 2022; Albergo et al., 2023), we further need to consider the specific characteristics of this problem; that is, the flow must transport the given $z_0$ to its *corresponding* $z_1$ (as defined in Eq. (8)), which is more complex than the ordinary flow matching that aims to learn transportation between two distributions in marginal level via *independent* sampling of $z_0$ and $z_1$. For example, our flow must map the latent of a specific yellow sofa to the latent of the *same* sofa rotated by a given angle, not to the rotated latent of a random object. As a result, we cannot sample $z_0$ and $z_1$ independently to construct the conditional path as usual, *i.e.*, $\pi_{0,1}(z_0, z_1) \neq p_0(z_0)p_1(z_1)$ in this problem.

A simple solution, without introducing extra overhead (*e.g.*, object-conditional embeddings), is strictly following the definitions of $z_0$ and $z_1$ in Eq. (8). Since $z_0 \sim p_0(z_0|x, g)$ and $z_1 \sim p_1(z_1|x, g)$, it is straightforward that the conditional joint distribution can be defined as $\pi_{0,1}(z_0, z_1|x, g) = p_0(z_0|x, g)p_1(z_1|x, g)$. Plugging this into Eq. (9) with additional marginalization over $(x, g) \sim q$ leads to

$$p_t(z_t) = \iiiint p_t(z_t|z_0, z_1) \, p_0(z_0|x, g) \, p_1(z_1|x, g) \, q(x, g) \, dx \, dg \, dz_0 \, dz_1. \tag{10}$$

Although Eq. (10) introduces four marginalizations, the terms over $z_0$ and $z_1$ collapse to Dirac measures in our case, since both are deterministically obtained from $(x, g)$ via the map $\Phi$ (Eq. (8)):

$$p_t(z_t) = \iiiint p_t(z_t|z_0, z_1) \, \delta\big(z_0 - \rho(g)\Phi(x)\big) \, \delta\big(z_1 - \Phi(g \circ x)\big) \, q(x, g) \, dx \, dg \, dz_0 \, dz_1 \tag{11}$$

$$= \iint p_t\big(z_t|\rho(g)\Phi(x), \, \Phi(g \circ x)\big) \, q(x, g) \, dx \, dg. \tag{12}$$

Following the flow matching framework, we construct the conditional probability path as a linear interpolation between $z_0 = \rho(g)\Phi(x)$ and $z_1 = \Phi(g \circ x)$:

$$p_t(z_t \mid z_0, z_1) = \mathcal{N}(z_t | (1-t)z_0 + tz_1, \sigma^2 I). \tag{13}$$

$\sigma$ is set to 0 to precisely satisfy the boundary conditions of $p_t$, leading to constant conditional velocity field over time (Liu et al., 2022): $v_t(z_t|z_0, z_1) = z_1 - z_0$. Then, our Residual Latent Flow (RLF) training objective is given by:

$$\mathcal{L}_{\text{RLF}}(\theta) = \mathbb{E}_{(x,g)\sim q(\cdot),\, t\sim\mathcal{U}[0,1],\, z_t\sim p_t(\cdot|z_0,z_1)} \left[ \|v_\theta(z_t, t) - (z_1 - z_0)\|_2^2 \right], \quad \text{where} \begin{cases} z_0 = \rho(g)\Phi(x) \\ z_1 = \Phi(g \circ x). \end{cases} \tag{14}$$

Upon convergence of the loss, flow $\hat{\psi}_1$ can be obtained by integrating the learned velocity field predictor $v_\theta$ from time 0 to 1, to transport $z_0$ to its corresponding $z_1$, *i.e.*, $\hat{\psi}_1(z_0) = \hat{z}_1 = z_0 + \int_0^1 v_\theta(z_\tau, \tau)\, d\tau$ (Fig. 3).

### 3.3 TRAINING DETAILS

**Baseline Autoencoder Training.** Our baseline implementation follows the original NFT setup (Koyama et al., 2023). We utilize ViT (Dosovitskiy et al., 2020) or U-Net (Ronneberger et al., 2015) for both encoder $\Phi$ and decoder $\Psi$. First, we train the autoencoder with the ERL loss (Eq. (2)) to get the latent reprsentations. Then, we freeze the encoder and train our flow model with our RLF loss (Eq. (14)) to correct the latents obtained from using the pre-trained encoder. We also attempted end-to-end training, but the process was unstable. We hypothesize that this instability stems from the latent distribution drifting as the autoencoder and flow model simultaneously updates.

**Decoder Fine-tuning.** After latent correction, the decoder $\Psi$ consumes flow-corrected latents, so we fine-tune $\Psi$ directly on that input distribution to remove the distribution shift between training and use. We update only $\Psi$ (freezing the encoder $\Phi$ and the flow model $\hat{\psi}_1$) by minimizing

$$\mathcal{L}_{\text{fine-tune,decoder}} = \mathbb{E}[\|g \circ x - \Psi(\mathtt{sg}(\hat{\psi}_1)(\rho(g)\mathtt{sg}(\Phi)(x)))\|_2^2]. \tag{15}$$

This objective adapts the decoder to its flow-corrected latent input distribution, aligning $\Psi$ to the corrected latent manifold while preserving supervision via the ground-truth image $g \circ x$. Freezing the encoder $\Phi$ and the flow model $\hat{\psi}_1$ prevents modifications that would undo the learned equivariance or the flow correction.

We fine-tune for 10–20% of the original training epochs with a same setting but reduced learning rate to 10% of the original learning rate. In practice this yields more faithful reconstructions and more better synthesis quality for both baseline and flow-based models. To guard against overfitting to local flow artifacts, we mix a small fraction of reconstruction loss, e.g., $\mathcal{L}_{\text{fine-tune,decoder}} + \lambda \mathbb{E}\left[\|x - \Psi(\Phi(x))\|_2^2\right]$ with $\lambda$ being a controllable parameter. Refer to D for details.

## 4 EXPERIMENTS

### 4.1 DATASETS

We evaluate our method on four datasets encompassing both geometric transformations $(SO(2), SO(3))$ and structured appearance changes. **ABO-Material** (Collins et al., 2022) and **ModelNet10-SO(3)** (Liao et al., 2019) are the $SO(3)$ cases; in ABO-Material, the object and its background co-rotate under viewpoint changes, whereas in ModelNet10-SO(3), the object rotates without background. **ComplexBRDFs** (Greff et al., 2022) is an $SO(2)$ setting, where the objects rotate only by a fixed axis. **ABO-Material Day-to-Night** is an $SO(2)$-style appearance variant where the object stays fixed and the illumination varies across 18 discrete angles at $10°$ intervals, producing day-night-like changes in shadows and colors. More details are provided in Sec. D.2.

### 4.2 EVALUATION METRICS

Let $x \in \mathbb{R}^{H \times W \times 3}$ be an input image and $g \in G$ a group element. We compare the performance of the following two models:

$$\hat{x}_{\text{base}}(g) := \Psi(\rho(g)\Phi(x)), \quad \hat{x}_{\text{ours}}(g) := \Psi\left(\hat{\psi}_{t=1}(\rho(g)\Phi(x))\right), \tag{16}$$

| GROUP | DATASET | METHOD | PRED ERROR↓ | PSNR↑ | LATENT ERROR↓ | ANGLE ERROR↓ |
|---|---|---|---|---|---|---|
| SO(3) | ABO | Base | $0.0667 \pm 0.0004$ | $11.85 \pm 0.01$ | $7.3 \times 10^{-4} \pm 1.3 \times 10^{-6}$ | $0.0079 \pm 0.0001$ |
| | | **Ours** | $\mathbf{0.0565} \pm 0.0004$ | $\mathbf{12.57} \pm 0.01$ | $\mathbf{1.9 \times 10^{-4}} \pm 5.1 \times 10^{-7}$ | $\mathbf{0.0010} \pm 0.0001$ |
| | ABO (OOD) | Base | $0.0641 \pm 0.0008$ | $12.14 \pm 0.01$ | $8.1 \times 10^{-4} \pm 1.7 \times 10^{-6}$ | $0.0088 \pm 0.0002$ |
| | | **Ours** | $\mathbf{0.0564} \pm 0.0008$ | $\mathbf{12.73} \pm 0.01$ | $\mathbf{2.1 \times 10^{-4}} \pm 7.6 \times 10^{-7}$ | $\mathbf{0.0012} \pm 0.0001$ |
| | ModelNet10-SO(3) (OOD) | Base | $0.1079 \pm 0.0025$ | $10.09 \pm 0.03$ | $7.2 \times 10^{-4} \pm 7.1 \times 10^{-6}$ | $0.1746 \pm 0.0064$ |
| | | **Ours** | $\mathbf{0.1018} \pm 0.0026$ | $\mathbf{10.43} \pm 0.03$ | $\mathbf{4.1 \times 10^{-4}} \pm 7.4 \times 10^{-6}$ | $\mathbf{0.0430} \pm 0.0018$ |
| | SmallNORB (OOD) | Base | $0.0052 \pm 0.0001$ | $23.13 \pm 0.02$ | $7.9 \times 10^{-5} \pm 2.7 \times 10^{-7}$ | $0.2429 \pm 0.0023$ |
| | | **Ours** | $\mathbf{0.0050} \pm 0.0001$ | $\mathbf{23.28} \pm 0.02$ | $\mathbf{4.8 \times 10^{-5}} \pm 2.6 \times 10^{-7}$ | $\mathbf{0.0728} \pm 0.0012$ |
| SO(2) | ABO Day-to-Night | Base | $0.0056 \pm 0.0001$ | $22.73 \pm 0.05$ | $1.9 \times 10^{-4} \pm 3.0 \times 10^{-6}$ | $0.0456 \pm 0.0006$ |
| | | **Ours** | $\mathbf{0.0039} \pm 0.0001$ | $\mathbf{24.32} \pm 0.05$ | $\mathbf{1.5 \times 10^{-4}} \pm 2.7 \times 10^{-6}$ | $\mathbf{0.0163} \pm 0.0003$ |
| | ABO Day-to-Night (OOD) | Base | $0.0079 \pm 0.0005$ | $21.80 \pm 0.08$ | $3.4 \times 10^{-4} \pm 6.6 \times 10^{-6}$ | $0.0776 \pm 0.0014$ |
| | | **Ours** | $\mathbf{0.0065} \pm 0.0005$ | $\mathbf{22.84} \pm 0.08$ | $\mathbf{2.6 \times 10^{-4}} \pm 6.0 \times 10^{-6}$ | $\mathbf{0.0269} \pm 0.0009$ |
| | ComplexBRDFs | Base | $0.0382 \pm 0.0009$ | $16.04 \pm 0.02$ | $8.9 \times 10^{-4} \pm 1.3 \times 10^{-5}$ | $0.0556 \pm 0.0003$ |
| | | **Ours** | $\mathbf{0.0296} \pm 0.0008$ | $\mathbf{17.38} \pm 0.02$ | $\mathbf{6.7 \times 10^{-4}} \pm 1.3 \times 10^{-5}$ | $\mathbf{0.0172} \pm 0.0004$ |
| | ComplexBRDFs (OOD) | Base | $0.0480 \pm 0.0026$ | $17.71 \pm 0.02$ | $1.4 \times 10^{-3} \pm 3.8 \times 10^{-5}$ | $0.0859 \pm 0.0009$ |
| | | **Ours** | $\mathbf{0.0404} \pm 0.0023$ | $\mathbf{19.19} \pm 0.03$ | $\mathbf{1.1 \times 10^{-3}} \pm 3.8 \times 10^{-5}$ | $\mathbf{0.0271} \pm 0.0010$ |
| | RotatedMNIST | Base | $0.0016 \pm 0.0000$ | $28.21 \pm 0.0061$ | $3.9 \times 10^{-5} \pm 1.4 \times 10^{-7}$ | $0.0032 \pm 0.0000$ |
| | | **Ours** | $\mathbf{0.0013} \pm 0.0000$ | $\mathbf{29.02} \pm 0.065$ | $\mathbf{3.7 \times 10^{-5}} \pm 1.4 \times 10^{-7}$ | $\mathbf{0.0030} \pm 0.0000$ |
| | RotatedMNIST (OOD) | Base | $0.0016 \pm 0.0000$ | $28.17 \pm 0.008$ | $3.9 \times 10^{-5} \pm 1.8 \times 10^{-7}$ | $0.0032 \pm 0.0000$ |
| | | **Ours** | $\mathbf{0.0013} \pm 0.0000$ | $\mathbf{28.99} \pm 0.009$ | $\mathbf{3.7 \times 10^{-5}} \pm 1.8 \times 10^{-7}$ | $\mathbf{0.0030} \pm 0.0000$ |

Table 1: **Comparison on novel view synthesis.** We report prediction error, PSNR, and latent error to measure the reconstruction quality and angle error to measure the violation strength of latent equivariance. Base indicates NFT Koyama et al. (2023). Note that SmallNORB and RotatedMNIST consist of real images.

where our Residual Latent Flow transports the analytic latent $\rho(g)\Phi(x)$ toward the target latent $\Phi(g \circ x)$. We use the following metrics to evaluate performance.

**Prediction Error.** We report the mean L2 distance between the synthesized image and the ground-truth transformed image in the pixel space $\|\hat{x}(g) - g \circ x\|_2^2$, where $\hat{x}(g) \in \{\hat{x}_{\text{base}}(g), \hat{x}_{\text{ours}}(g)\}$ and the ground-truth transformed image is denoted as $g \circ x$.

**Peak Signal-to-Noise Ratio (PSNR).** Computed between the predicted image $\hat{x}(g) \in \{\hat{x}_{\text{base}}(g), \hat{x}_{\text{ours}}(g)\}$ and the ground-truth transformed image $g \circ x$, it quantifies the reconstruction fidelity in the pixel space using a logarithmic scale. A higher PSNR value indicates superior low-level image quality and less distortion from the true transformed view.

**Latent Error.** Similarly, we report the L2 distance between the predicted rotated latent and the latent of ground-truth transformed image in the latent space $\|\Psi^{-1}\hat{x}(g) - \Phi(g \circ x)\|_2^2$, where $\hat{x}(g) \in \{\hat{x}_{\text{base}}(g), \hat{x}_{\text{ours}}(g)\}$ and the latent of ground-truth transformed image is denoted as $\Phi(g \circ x)$.

**Angle Error.** To measure the strength of violation of latent equivariance, we compare the predicted latent $\Phi(g \circ x)$ with the analytically rotated source latent $\rho(g)\Phi(x)$.

For SO(2), for each degree-$\ell$ block from both $\Phi(g_\theta \circ x)$ and $\rho(g_\theta)\Phi(x)$, we estimate the angle $\hat{\delta\theta}_\ell$ that minimizes their Frobenius norm discrepancy:

$$\hat{\delta\theta}_\ell = \arg\min_{\delta\theta} \|\Phi_\ell(g_\theta \circ x) - R_\ell(\delta\theta)\,\Phi_\ell(x)\|_F^2, \tag{17}$$

where $R_\ell(\theta)$ denotes the degree-$\ell$ block of $\rho(g_\theta)$ and $\Phi_\ell(\cdot)$ denotes the degree-$\ell$ representation. Higher-degree SO(2) blocks exhibit degenerate angle solutions, and we select the smallest magnitude of each $\hat{\delta\theta}_\ell$ as a representative. Then, we take the average over the degrees of representations $\hat{\delta\theta} = \frac{1}{L}\sum_{\ell=1}^{L}\hat{\delta\theta}_\ell$ and take the angular distance as a metric to measure the accuracy: $d_{\cos}(\hat{\delta\theta}) = 1 - \cos(\hat{\delta\theta})$.

For SO(3), only the degree-1 block is invertible with respect to the underlying SO(3) rotation parameters, whereas higher-degree Wigner $D$ blocks provide equivariant embeddings that do not uniquely

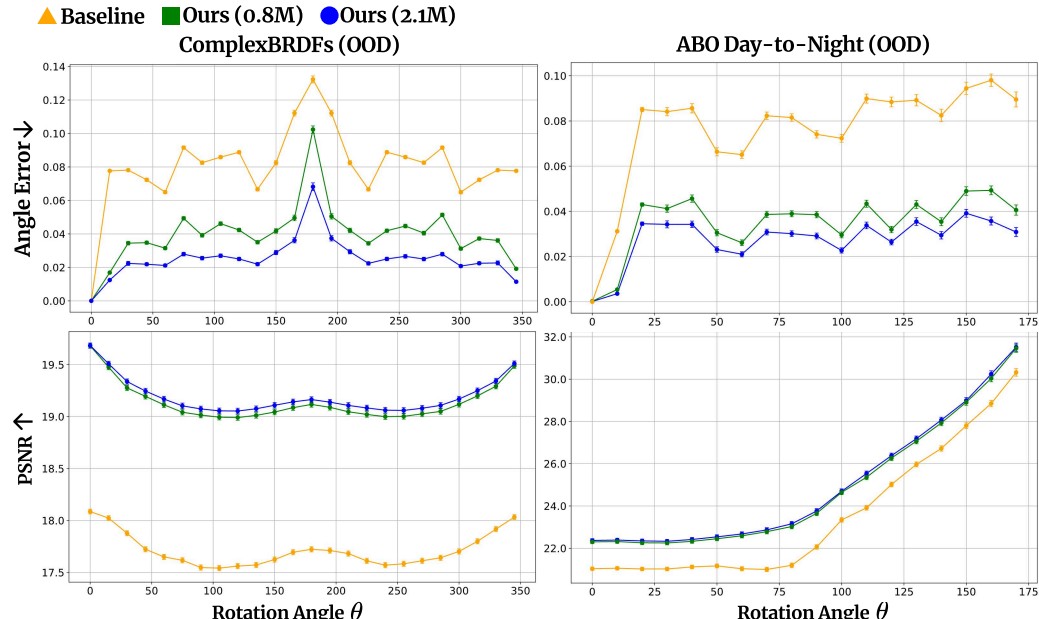

Figure 4: **Quantitative evaluation on SO(2) datasets across angular displacements**. The results are shown for two flow backbones with 0.8M and 2.1M parameters. Rotation angle $\theta$ denotes the angular displacement (degrees) applied to every object/viewpoint. **Top:** Angle error as a function of rotation angle. **Bottom: PSNR.** These are evaluated on 57,360 pairs in ComplexBRDFs OOD set and on 6,732 pairs in ABO Day-to-Night OOD set.

determine $(\alpha, \beta, \gamma)$. Therefore, we use the degree-1 block to numerically solve the Wahba problem (Markley, 1988) (detailed in Sec. C), estimating the optimal rotation $\hat{R}(\alpha, \beta, \gamma) \in \mathrm{SO}(3)$ that aligns $\Phi(g_{\alpha,\beta,\gamma} \circ x)$ and $\rho(g_{\alpha,\beta,\gamma})\Phi(x)$:

$$\hat{R}(\alpha, \beta, \gamma) = \underset{R \in \mathrm{SO}(3)}{\arg\min} \|\Phi_{\ell=1}(g_{\alpha,\beta,\gamma} \circ x) - R(\alpha, \beta, \gamma)\,\Phi_{\ell=1}(x)\|_F^2\,, \tag{18}$$

where we denote by $R(g)$ the degree-1 block of $\rho(g)$. For comparing higher degree blocks, we provide latent error, which directly computes L2 distance between the predicted latents and ground-truth latents, for all higher degree representations. For perfect equivariance, the relative rotation between $\Phi(g \circ x)$ and $\rho(g)\Phi(x)$ should be the identity rotation. To measure the accuracy, the solution $\hat{R}$ is converted to a quaternion $\hat{q}$ and then we compute the cosine distance from the identity quaternion:

$$d_{\cos}(\hat{q}, q_{\mathrm{id}}) = 1 - |\langle \hat{q}, q_{\mathrm{id}} \rangle|, \quad q_{\mathrm{id}} = (1, 0, 0, 0). \tag{19}$$

### 4.3 COMPARISON ON NOVEL VIEW SYNTHESIS

#### 4.3.1 IN-PLANE ROTATION SYNTHESIS

| Method | PSNR ↑ | Pred Error ↓ | LPIPS ↓ | SSIM ↑ |
|---|---|---|---|---|
| SpatialVAE (Bepler et al., 2019) | 15.59 | 0.0373 | 0.1561 | 0.6664 |
| GIAE (Winter et al., 2022) | 19.88 | 0.0146 | 0.1000 | 0.8289 |
| LGA (Jin et al., 2024) | 24.13 | 0.0049 | 0.0385 | 0.9427 |
| NFT (Koyama et al., 2023) | 28.21 | 0.0016 | 0.0035 | 0.9953 |
| **Ours** | **29.02** | **0.0013** | **0.0030** | **0.9961** |

Table 2: **Comparison on in-plane rotation NVS using RotatedMNIST (SO(2)).**

First, we evaluate whether our method is effective for in-plane rotation synthesis, a special case of NVS where the rotation axis is parallel to the viewing direction, *e.g.*, digits rotating within the same plane in RotatedMNIST. In Tab. 2, we show comparison between Spatial-VAE (Bepler et al., 2019),

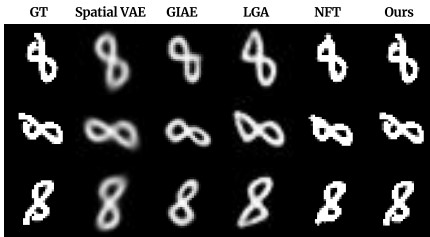

Figure 5: Qualitative comparison on in-plane rotation NVS using RotatedMNIST (SO(2)).

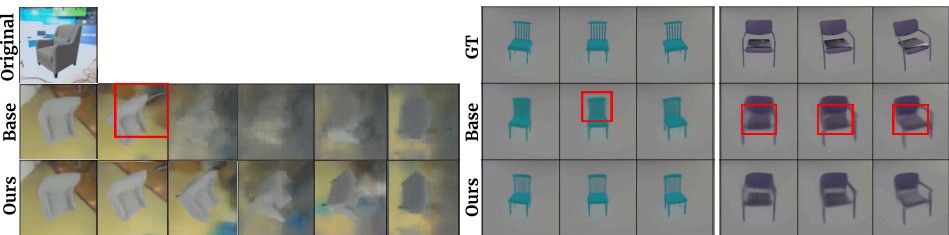

Figure 6: **Qualitative comparison on out-of-plane NVS** (SO(3)) **on OOD datasets. Left:** Results from the ABO-Material (OOD) with test-time SO(3) rotations without ground-truth. Base indicates NFT Koyama et al. (2023). As the rotation angle grows, the baseline exhibits more corrupted renderings where the background is not well preserved. **Right:** Results from the ComplexBRDFs (OOD) with SO(2) rotations. Our method retains structural fidelity, capturing the fine details of the original image.

GIAE (Shakerinava et al., 2022), LGA (Jin et al., 2024), NFT (Koyama et al., 2023), and ours. In Fig. 5, we provide qualitative examples. This result corroborates that our latent correction method is effective and outperforms the existing baselines.

### 4.3.2 Out-of-plane Rotation Synthesis

| Method | PSNR ↑ | Pred Error ↓ | LPIPS ↓ | SSIM ↑ |
|---|---|---|---|---|
| LGA (Jin et al., 2024) | 17.59 | 0.0174 | 0.270 | 0.785 |
| ENR (Dupont et al., 2020) | 18.07 | 0.0156 | 0.262 | 0.800 |
| NFT (Koyama et al., 2023) | 23.13 | 0.0052 | 0.272 | **0.811** |
| **Ours** | **23.28** | **0.0050** | **0.247** | 0.802 |

Table 3: **Comparison on out-of-plane rotation NVS using SmallNORB (**SO(3)**).**

Out-of-plane rotation synthesis is another form of novel view synthesis, but in a more challenging setting: the object rotates around an axis that is not parallel (typically orthogonal) to the viewing direction. In this case, the model has to infer the appearance of occluded parts, making the task significantly harder.

In Tab. 1 and Tab. 3, we present a comprehensive comparison across various datasets that has SO(2) or SO(3) group symmetry. We evaluate performance under two conditions: *(i)* out-of-distribution (OOD) objects, which were never seen during training, and *(ii)* in-distribution objects, which were seen during training but presented at unseen test-time angles. On ModelNet10-SO(3) and small-NORB dataset, we evaluate only on unseen objects (OOD) as no dedicated test split is available. Our method shows consistent improvements in both of unseen objects and unseen angles, strongly indicating that our method effectively corrects the misaligned latents to faithfully decodable true latents and thus enhances the fidelity of decoded images.

Additionally, as shown in Fig. 4, our method achieves stable improvements in all evaluation metrics regardless of the specific rotation angle, indicating that our loss objective Eq. (14) is effectively

designed for jointly coupled boundary distributions. Specifically, two main key components of our objective are: *(i)* $\psi_1$ transport given source point to *corresponding* target point, not to the arbitrary point of the marginal target distribution, and *(ii)* each target point should be mapped from various source points with arbitrary rotation angles. Improved performance over various datasets (Tab. 1) shows the effectiveness of the first component, while the improvement over all rotation angles (Fig. 4) validates the second claim. Note that, we show the scalability of model size for our method, comparing two flow model variants with different capacities (0.8M vs. 2.1M). The larger model yields the strongest performance, but even the smaller model consistently outperforms the baseline.

Moreover, we visualize the qualitative examples in Fig. 6, which further supports the effectiveness of our method. The rotation angle increases from left to right in each example. More qualitative examples can be found in Sec. J.

## 5 RELATED WORK

**Group Symmetry-aware Models.** Equivariant neural networks and Lie group-informed models (Falorsi et al., 2018; Quessard et al., 2020; Shakerinava et al., 2022; Hayashi et al., 2025; Bertolini et al., 2025) explicitly incorporate symmetry constraints into neural architectures. Notable studies include group-equivariant convolutional networks (Cohen & Welling, 2016), homeomorphic variational auto-encoders leveraging Lie group symmetries (Falorsi et al., 2018), and other frameworks employing group structures to enhance interpretability and robustness, including steerable and topographic parameterizations (Bökman et al., 2024; Keller & Welling, 2021), symmetry discovery and transformation-aware representations (Hinton et al., 2011; Cohen & Welling, 2014; Park et al., 2022; Miyato et al., 2022), and equivariant latent modeling (Dupont et al., 2020; Song et al., 2023a).

Recently, group symmetry-based diffusion and flow matching models have successfully tackled challenges across various domains, including molecular generation (Hoogeboom et al., 2022; Guan et al., 2023; Song et al., 2023b), and robotics (Ryu et al., 2024; Wang et al., 2024; Braun et al., 2024).

**Geometry-aware Diffusion and Flow Matching.** Diffusion and Flow matching (Lipman et al., 2022; Liu et al., 2022; Albergo et al., 2023) integrated with geometric modeling is an emerging area of research. These methods improve controllability and interpretability of the generative model (Hahm et al., 2024). Recent work demonstrates Diffusion and Flow matching on general geometry (Chen & Lipman, 2023; Sherry & Smets, 2025; De Bortoli et al., 2022), manifolds (Kapusniak et al., 2024), and equivariant distributions and vector fields (Kim et al., 2025; Wang et al., 2025).

**Generative Models for Novel View Synthesis.** Generative frameworks (Ho et al., 2020; Song et al., 2020; Goodfellow et al., 2020) have notably advanced novel view synthesis (NVS), prominently exemplified by NeRF-based models (Mildenhall et al., 2021; Lin et al., 2023; Yu et al., 2021). Recent specialized generative models targeting NVS have achieved notable improvements in visual quality and view consistency (Karnewar et al., 2023; Yu et al., 2023; Shi et al., 2023; Anciukevičius et al., 2023; Ye et al., 2024; Chan et al., 2022).

## 6 SUMMARY

In this work, we present a flow-based latent correction framework to address the challenge of misalignment in equivariant models. By learning flow to transport analytically rotated latents toward their empirically encoded targets, our method restores geometric consistency while preserving the inductive biases of group symmetries. Empirical studies on rotation groups and structured appearance transformations demonstrate substantial gains in both alignment fidelity and reconstruction quality.

As a limitation, our flow correction relocates latents toward the target manifold but final image quality is upper bounded by the capacity of the decoder. One promising future direction would be to raise the ceiling of the decoder using a higher capacity or multi–scale architecture, adding perceptual losses, or developing a hierarchical encoder and decoder based equivariant representation learning on Diffusion or Flow Matching framework.

ETHICS STATEMENT

Our work focuses on developing computational methods for geometry-aware flow matching and novel view synthesis. The techniques are purely theoretical and computational, relying exclusively on synthetic image datasets with controlled geometric variations. No human subjects, personal data, or sensitive contents are involved. We therefore identify no ethical concerns arising from this research.

REPRODUCIBILITY STATEMENT

We provide complete details of our experimental setup, datasets, and model architectures and hyperparameters in Section D. Descriptions of both Residual Latent Flow correction framework and the procedures for generating synthetic rotational image datasets are given in the main text. Source code will be released upon acceptance to ensure full reproducibility.

USE OF LLMS

Large language models were used solely to refine grammar and phrasing during manuscript preparation.

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

# APPENDIX

## A CLASSES OF NOVEL VIEW SYNTHESIS

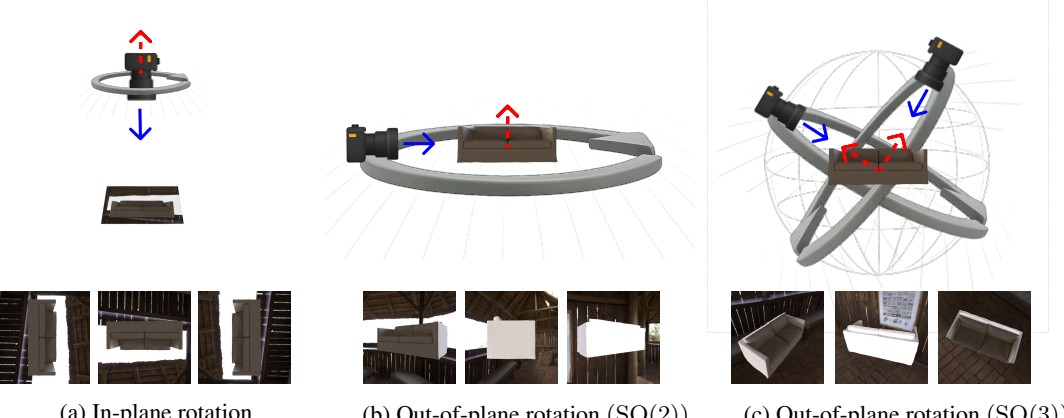

(a) In-plane rotation      (b) Out-of-plane rotation $(SO(2))$      (c) Out-of-plane rotation $(SO(3))$

Figure I: **Illustration depicting different classes of Novel View Synthesis (NVS) tasks.** Blue arrow indicates camera's viewing direction and red arrow indicates the axis of rotation. (a) In-plane rotation: the camera rotates around an axis parallel to the viewing direction. (b) Out-of-plane rotation with $SO(2)$ freedom: the camera rotates around an axis that is not parallel to the viewing direction. (c) Out-of-plane rotation with full $SO(3)$ freedom: the camera can move in an arbitrary direction in 3D space, representing the most challenging NVS scenario.

Novel View Synthesis (NVS) (Miyato et al., 2022; Koyama et al., 2023; Miyato et al., 2023) is a task to generate realistic images of a specific subject or scene from a specific point of view, given a set of images for the same scene taken from different viewpoints. The NVS task can be categorized into different classes by the types of allowed rotation of the object (or equivalently, the camera viewpoint). Fig. I illustrates several representative classes of the NVS tasks. The in-plane rotation shown in (a) allows the object (or the camera) to rotate around an axis that is parallel to its viewing direction. Since this is equivalent to rotating the resulting images in the same 2D space, this is a relatively easy task. Out-of-plane rotations, on the other hand, occur when the camera view and rotation axis are not aligned. In particular, out-of-plane rotation under $SO(2)$ freedom shown in (b) allows the rotation axis not to be in parallel to the viewing direction. Thus, the area and shape of the object in the projected 2D image varies by the rotation angle, making it significantly more challenging compared to the in-plane rotation. The most challenging scenario is shown in (c), where the camera can move in the 3D space both latitudinally and longitudinally at the same time under rotational freedom of $SO(3)$, requiring the model to predict views from arbitrary angles. In this paper, we aim to tackle all of these challenging NVS task, not just the simple in-plane rotation problem.

## B WIGNER $D$-MATRIX REPRESENTATION

**Conventions and indices.** We use the active Z–Y–Z Euler convention for $g \in SO(3)$ with angles $(\alpha, \beta, \gamma)$. Fix an integer degree $\ell \in \{0, 1, 2, \dots\}$ and the index set $\mathcal{I}_\ell = \{-\ell, \dots, \ell\}$. All $(2\ell+1) \times (2\ell+1)$ matrices below are written in the spherical basis ordered by rows/columns $m, n \in \mathcal{I}_\ell$. The generators of $SO(3)$, $J_x^{(\ell)}, J_y^{(\ell)}, J_z^{(\ell)}$, (Hermitian matrices acting on this space) satisfy $[J_x, J_y] = iJ_z$ and cyclic permutations.

**Definition.** The Wigner $D$-matrix is the matrix exponential product
$$D^{(\ell)}(g) = \exp(-i\alpha J_z) \exp(-i\beta J_y) \exp(-i\gamma J_z) \in \mathbb{C}^{(2\ell+1)\times(2\ell+1)},$$
with entries $D_{m,n}^{(\ell)}(g) = [D^{(\ell)}(g)]_{mn}$. In Z–Y–Z convention, $D^{(\ell)}$ factorizes elementwise as
$$D_{m,n}^{(\ell)}(\alpha, \beta, \gamma) = e^{-im\alpha} d_{m,n}^{(\ell)}(\beta) e^{-in\gamma},$$

and the *small-d* matrix $d^{(\ell)}(\beta) = \exp(-i\beta J_y)$ is real.

**Closed form for the small-$d$ entries.**  With row/column order $(m, n)$,

$$d^{(\ell)}_{m,n}(\beta) = \sqrt{(\ell+m)!\,(\ell-m)!\,(\ell+n)!\,(\ell-n)!} \sum_{s=s_{\min}}^{s_{\max}} \frac{(-1)^{m-n+s}\,(\cos\frac{\beta}{2})^{2\ell+n-m-2s}\,(\sin\frac{\beta}{2})^{m-n+2s}}{(\ell+n-s)!\,s!\,(m-n+s)!\,(\ell-m-s)!},$$

where $s_{\min} = \max(0, n-m)$ and $s_{\max} = \min(\ell+n, \ell-m)$.

**Real basis and $\ell$=1 block.**  A fixed change of basis $Q_\ell$ gives $D^{(\ell)}_{\mathrm{real}}(g) = Q_\ell D^{(\ell)}(g) Q_\ell^\top$ (real orthogonal). For $\ell = 1$, $D^{(1)}_{\mathrm{real}}(g)$ equals the standard $3 \times 3$ rotation $R(g)$.

If we restrict to SO(2) rotations (i.e. $\beta = \gamma = 0$ and $\alpha = \theta$), the Wigner $D$-matrices reduces to

$$D^\ell_{m,n}(\theta, 0, 0) = \delta_{m,n}\,e^{-im\theta}. \tag{20}$$

Thus the $(2\ell + 1)$-dimensional irrep of SO(3) decomposes under restriction as

$$D^\ell(\theta)\Big|_{\mathrm{SO}(2)} \cong \rho_{-\ell}(\theta) \oplus \rho_{-\ell+1}(\theta) \oplus \cdots \oplus \rho_\ell(\theta). \tag{21}$$

In a real basis, the complex conjugate pair $\rho_m$ and $\rho_{-m}$ can be combined into a $2 \times 2$ rotation block:

$$R_m(\theta) = \begin{pmatrix} \cos(m\theta) & -\sin(m\theta) \\ \sin(m\theta) & \cos(m\theta) \end{pmatrix}, \qquad m \geq 1, \tag{22}$$

together with the trivial representation $R_0(\theta) = [1]$. We will use the direct product of these real representations $\phi(g) = \bigoplus_{m=0}^{n-1} \phi_m(g)$ as the group representation of SO(2).

**Connection to our model.**  Our representation decomposes as

$$B(g) = P\,\rho(g)\,P^{-1} = \bigoplus_{\ell=0}^{L} \left(D^{(\ell)}(g) \otimes I_{m_\ell}\right),$$

so the $\ell$-th latent block transforms exactly by $D^{(\ell)}(g)$ (or its real form); we exploit the $\ell = 1$ block for the Wahba alignment in the main text.

## C   WAHBA PROBLEM

Given weighted direction correspondences $\{(r_i, b_i, a_i)\}_{i=1}^n$ with $a_i > 0$ and $\sum_i a_i = 1$, Wahba's problem seeks the proper special orthogonal matrix (rotation matrix) $A \in \mathrm{SO}(3)$ minimizing

$$L(A) = \tfrac{1}{2}\sum_{i=1}^n a_i\,\|b_i - Ar_i\|^2 = 1 - \mathrm{tr}(AB^\top), \qquad B = \sum_{i=1}^n a_i\,b_i r_i^\top. \tag{23}$$

Let the singular value decomposition be $B = U\,S\,V^\top$ with $S = \mathrm{diag}(s_1, s_2, s_3)$, $s_1 \geq s_2 \geq s_3 \geq 0$. Define $d = \det(U)\det(V) \in \{+1, -1\}$. Then Markley's SVD solution gives

$$A_{\mathrm{opt}} = U\,\mathrm{diag}(1, 1, d)\,V^\top \in \mathrm{SO}(3), \tag{24}$$

which minimizes $L(A)$. Intuitively, writing $W = U^\top AV$ yields $L(A) = 1 - \mathrm{tr}(S'W)$ with $S' = \mathrm{diag}(s_1, s_2, d\,s_3)$, and the minimum occurs at $W = I$.

**Properties and uniqueness.**  If $\mathrm{rank}(B) \geq 2$ (i.e., $s_2 > 0$), the solution is unique except in the degenerate limit where $B$ is near rank $< 2$; in that case a one-parameter family of minimizers appears (rotation about an axis). The SVD approach is numerically robust (avoids squaring $B$) and, unlike certain fast implementations of Davenport's $q$-method, naturally exposes eigen-structure used for covariance analysis; see Markley (1988) for closed-form covariance expressions.

This is a procedure for solving the Wahba problem:

1. Form $B = \sum_i a_i b_i r_i^\top$ (normalize $\sum_i a_i = 1$).

2. Compute $B = USV^\top$

3. Set $d = \det(U)\det(V)$.

4. Return $A_{\mathrm{opt}} = U\,\mathrm{diag}(1, 1, d)\,V^\top$ (guarantees $\det(A_{\mathrm{opt}}) = +1$).

# D  IMPLEMENTATION DETAILS

## D.1  TRAINING DETAILS

We train all flow models for 300 epochs. We use AdamW optimizer with batch size of 128, learning rate of $10^{-4}$, weight decay $0.05$. The encoder is ought to be optimized jointly across all blocks, to be able to capture empirical misalignment between analytic and learned latents introduces dependencies among them. This motivates the use of architectures capable of capturing inter-block correlations. For $SO(2)$, we used 4 layers of multi-head self attention based Transformer encoder layers with 128 (0.8M parameters) and 256 (2.1M parameters) latent's channel dimension, 8 heads and 512 hidden dimension. For $SO(3)$, we used U-net based structure with 4 residual blocks, 256 channels, 8 heads and 64 hidden dimension.

## D.2  DATASET

**ABO-Material.** This dataset (Collins et al., 2022) consists of rendered images from the Amazon-Berkeley Objects (ABO) collection. Each object is rendered from 91 viewpoints uniformly distributed over the upper hemisphere of an icosphere, with variations in azimuth and elevation. To introduce contextual diversity, each object is rendered under three distinct high-dynamic-range (HDR) environment maps with different lighting and backgrounds.

**ModelNet10-SO(3).** This dataset (Liao et al., 2019) contains clean, object-centric renderings of CAD models from the ModelNet10-SO(3) benchmark. Objects are viewed from random $SO(3)$ rotations without background or environmental textures, providing a minimal setting for analyzing pure rotational equivariance.

**ComplexBRDFs.** This dataset (Greff et al., 2022) comprises object-only renderings of ShapeNet models under complex materials (e.g., metallic, glossy). Each object undergoes a full 360° in-plane rotation (about the z-axis), sampled at 24 evenly spaced steps. This yields a structured $SO(2)$ transformation setting, focusing on view consistency under material-induced appearance variation.

**ABO-Material Day-to-Night.** To evaluate generalization beyond rigid geometric transformations, we introduce a variant of the ABO-Material dataset (Collins et al., 2022). Each object is rendered from a fixed viewpoint while lighting direction changes along a 170° arc in 10° increments. This results in 18 lighting conditions per object, simulating a structured day-to-night transition with shadows and specular variation. This defines a quasi-$SO(2)$ transformation in appearance space, without explicit rotation of object geometry.

**RotatedMNIST.** RotatedMNIST is a planar rotation variant of the MNIST dataset (Deng, 2012), where each digit image is transformed by an in-plane $SO(2)$ rotation. In the standard release, the dataset provides pre-rotated images, with each sample rotated by a uniformly sampled angle in $[0, 2\pi)$. This forms a clean benchmark for evaluating equivariance to 2D rotational transformations under controlled image statistics.

**SmallNORB.** The SmallNORB dataset (LeCun et al., 2004) contains images of physical toy objects captured using a real camera under controlled conditions. Each object instance is photographed across 18 azimuth angles, 9 elevations, and 6 lighting directions, producing systematic variations in viewpoint and illumination. As the images stem from actual camera captures rather than synthetic renderings, the dataset provides a realistic benchmark for testing equivariance and robustness to real-world visual factors.

**Shape of latent representations.** The latent representation shapes depend on the underlying symmetry group: for $SO(3)$, the latents have a shape of $C \times 81$, corresponding to the block-diagonal Wigner-$D$ representation with degrees $\ell = 0, \ldots, 8$. We use $C = 128$ for ABO-Material and small-NORB, and $C = 64$ for ModelNet10-SO(3). For processing with a U-Net, the latents are reshaped to $C \times 9 \times 9$. For $SO(2)$, these latents have a shape of $C \times 17$. We use $C = 128$ for ComplexBRDFs and ABO-Material Day-to-Night, and $C = 64$ for RotatedMNIST.

# E    TRANSFER TO ROTATION ESTIMATION

| Method | Test Error | OOD Error |
|--------|-----------|-----------|
| NFT | 0.002507 | 0.004473 |
| **Ours** | **0.001131** | **0.001961** |

Table I: **Rotation estimation on ABO Day-to-Night.**

To test whether the latent representations that better preserve the underlying rotational structure improves the performance in other downstream tasks such as pose estimation, we compare the performance on the rotation angle prediction task using ABO Day-to-Night.

Given a latent feature tensor $z \in \mathbb{R}^{64 \times 17}$ produced by the encoder, we attach a light regression head consisting of two fully-connected layers with ReLU activations, followed by a linear output layer that predicts a rotation angle $\theta_0$ of the given image. We only train the regression head while keeping the latent encoder frozen.

For the NFT baseline, the latent rotation is obtained analytically through the group action $D(\Delta\theta)z$, and the regressor is trained to recover the target angle $\theta_1 = \theta_0 + \Delta\theta$. For our method, the analytically rotated latent is further refined through the learned flow module before angle prediction. We measure accuracy using the cosine-based angular discrepancy $1 - \cos(\hat{\theta}_1 - \theta_1)$.

Tab. I compares the rotation angle estimation performance of our method and that of the NFT baseline. We observe that our method achieves significantly lower angular discrepancy on both the test and OOD sets, verifying that our approach is applicable to this different downstream task.

# F    ABLATION ON NOISE LEVEL OF STOCHASTIC PATH

| Stochasticity ($\sigma$) | Latent Error | PSNR | Prediction Error |
|--------------------------|--------------|------|------------------|
| **0** | **0.0002607** | **22.84** | **0.0065** |
| 0.01 | 0.0002872 | 22.68 | 0.0067 |
| 0.05 | 0.0003631 | 22.22 | 0.0074 |
| 0.1 | 0.0005406 | 19.30 | 0.0129 |

Table II: **Stochastic Interpolation Results with Varying Noise Levels.**

To construct a stochastic interpolant (Albergo et al., 2023), we add a stochastic correction to the velocity instead of injecting noise into the state. Let $x_0$ and $x_1$ be the endpoints. The deterministic linear interpolant is $\mu_t = (1 - t)x_0 + tx_1$, representing the mean trajectory, with velocity $v_{\text{det}} = x_1 - x_0$.

The stochastic correction, derived from the Brownian bridge drift, keeps the trajectory anchored at the endpoints:

$$v_{\text{sto}}(t) = \frac{1 - 2t}{2t(1 - t)}(x_t - \mu_t).$$

The full velocity is $v(t) = v_{\text{det}} + \sigma\, v_{\text{sto}}(t)$, where $\sigma$ controls stochastic strength. Setting $\sigma = 0$ recovers the deterministic interpolant, while larger $\sigma$ increases trajectory variability.

In our ablation study, we vary $\sigma \in \{0, 0.01, 0.05, 0.1\}$ and measure latent reconstruction error, PSNR, and prediction error. Our ablation results over the noise scale in Tab. II indicate that a larger stochasticity gradually degrades latent reconstruction, PSNR, and prediction accuracy. In other words, since our objective is to learn a precise deterministic correction path rather than to model a high–entropy family of trajectories, strong stochastic perturbations of the interpolant are not beneficial in this regime and can even hinder optimization.

Taking a deeper look, under our setting, the flow is used purely as a deterministic correction map from a misaligned latent $z_0$ to its aligned counterpart $z_1$. Thus, the underlying conditional distribution $p(z_1 \mid z_0)$ is effectively low–entropy and close to a one-to-one mapping. While the stochastic interpolant framework of Albergo et al. (2023) allows one to introduce nontrivial diffusion along the path without changing the endpoint marginals, this additional stochasticity does not enrich the target distribution in our correction scenario. Instead, increasing the noise level merely enlarges the variance of the training trajectories $x_t$ around the same endpoints, which acts as label noise for the velocity field.

## G  EXPERIMENT ON SCALABILITY

| $C$ | Method | PSNR | Latent Error | Pred Error | Latency (ms/sample) | Params (M) |
|---|---|---|---|---|---|---|
| 32 | Base | 20.20 | 0.00008 | 0.0100 | 0.969 | 10 |
| | Ours | 20.56 | 0.00006 | 0.0092 | 1.265 | 10 + 2 |
| 64 | Base | 20.79 | 0.00013 | 0.0087 | 0.979 | 32 |
| | Ours | 21.12 | 0.00010 | 0.0081 | 1.310 | 32 + 2 |
| 128 | Base | 22.73 | 0.00019 | 0.0056 | 1.298 | 120 |
| | Ours | 24.32 | 0.00015 | 0.0039 | 1.785 | 120 + 2 |

Table III: **Performance as latent channel dimension $C$ varies with fixed max degree $L = 8$.**

| $L$ | Method | PSNR | Latent Error | Pred Error | Latency (ms/sample) | Params (M) |
|---|---|---|---|---|---|---|
| 2 | Base | 22.34 | 0.00009 | 0.0061 | 1.155 | 101 |
| | Ours | 22.73 | 0.00008 | 0.0056 | 1.743 | 101 + 2 |
| 4 | Base | 22.48 | 0.00008 | 0.0060 | 1.179 | 107 |
| | Ours | 22.99 | 0.00007 | 0.0053 | 1.758 | 107 + 2 |
| 8 | Base | 22.73 | 0.00019 | 0.0056 | 1.298 | 120 |
| | Ours | 24.32 | 0.00015 | 0.0039 | 1.785 | 120 + 2 |

Table IV: **Performance as maximum representation degree $L$ varies with fixed latent channel $C = 128$.**

**Inference Time Measurement.** All models are benchmarked on a single NVIDIA RTX A6000 GPU in `eval` mode with all parameters frozen. For each configuration, we perform one warm-up validation pass and then measure the runtime of a second `validate()` call, placing `torch.cuda.synchronize()` before and after the measurement for accurate GPU timing. The per-sample latency is obtained by dividing the total validation time by the number of samples in the OOD split. For Flow models, latent correction was performed using a fixed 10-step generation procedure.

**Flow Matching Training Cost.** To quantify the computational overhead of our flow module, we additionally summarize its training-time characteristics. On the SO(2) RotMNIST setting, the flow model runs at approximately 65ms per optimization step with batch size 256 and uses about 1.96GB of GPU memory on a single NVIDIA RTX A6000. On the SO(3) ABO setting, the corresponding flow model takes roughly 110ms per step with the same batch size, and consumes around 2.15GB of GPU memory.

## H  EXPERIMENT UNDER NOISY LABEL SETTING

We corrupt the labels by up to $k^\circ$, with $k \in \{2.5, 5.0, 7.5, 10.0\}$ on the 10% of the training samples. For each noise level, we report PSNR on the OOD and test splits in Tab. V, comparing with the NTF

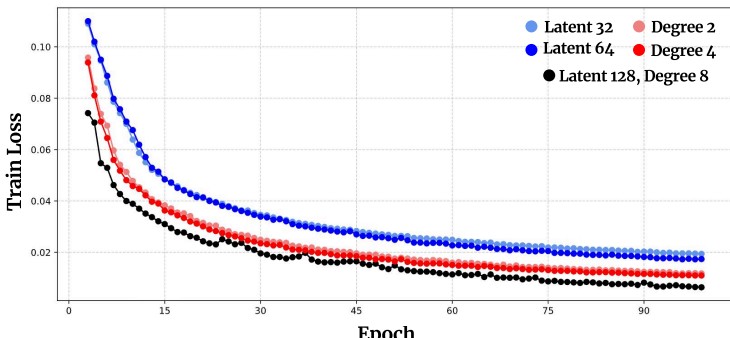

Figure II: **Experiment on Scaling.** The NFT's train loss converges stably across different latent scales, demonstrating that the NFT baseline remains reliable under varying dimensional settings.

| Noise level | Split | NFT (PSNR) | Ours (PSNR) |
|---|---|---|---|
| $10.0°$ | OOD | 17.39 | 17.52 |
| $10.0°$ | Test | 17.82 | 17.95 |
| $7.5°$ | OOD | 18.81 | 19.08 |
| $7.5°$ | Test | 19.27 | 19.56 |
| $5.0°$ | OOD | 20.60 | 20.90 |
| $5.0°$ | Test | 21.17 | 21.53 |
| $2.5°$ | OOD | 21.40 | 21.83 |
| $2.5°$ | Test | 21.85 | 22.56 |
| $0°$ | OOD | 21.80 | 22.84 |
| $0°$ | Test | 22.73 | 24.32 |

Table V: **Effect of noisy rotation labels on the ABO-Material Day-to-Night dataset.**

baseline with our flow-corrected model. We observe that our method consistently outperforms the NFT baseline both on the OOD and test split across all the tried noise levels. Also, the performance degradation is mild enough to use in practice, unless the noise level is relatively high (*e.g.*, larger than $10°$).

## I  MOTIVATION: CONFLICTING LOSS OBJECTIVES IN ERL

The equivariance loss enforces the latent of a transformed image $\Phi(g \circ x)$ to match the analytically rotated latent $\rho(g)\Phi(x)$, thereby promoting equivariance of the encoder. At the same time, the decoder loss optimizes autoencoder for precise reconstruction by encouraging the decoder to decode latent $\Psi(\rho(g)\Phi(x))$ to match the ground truth of the transformed image $g \circ x$.

Ideally, if the encoder were perfectly trained, the encoded latent would behave well-aligned with the group actions, where latent trajectories along the angle (latent trajectory) under linear transformations align cleanly along a fixed orbit $\text{Orb}(x) := \{g \circ x \mid g \in G\}$ generated by the defined group actions. As illustrated in the left sphere of Fig. 2a, the degree-1 representations on $\text{SO}(3)$ would ideally trace smooth circular path confined to the surface of the sphere, such that trajectories originating from different starting viewpoints converge to the same target view. In practice, however, as shown in the right sphere of Fig. 2a, the paths deviate from the ideal spherical orbit. The latents scatter off the surface, and applying transformation from different viewpoints no longer leads to a consistent target.

While the encoder should preserve the structure of group transformations via $\mathcal{L}_{\text{equiv}}$, it also needs to encode sufficient object detail (*e.g.*, texture, shape) to enable faithful image reconstruction to minimize $\mathcal{L}_{\text{decoder}}$. For example, consider a set of images depicting a perfectly uniform sphere with uniform lighting. Any rotation about the $z$-axis results in visually indistinguishable images. From a reconstruction standpoint, the encoder would assign an identical latent representation for every view.

Meanwhile, to satisfy the equivariance relation, the model must encode every viewpoints differently, as they correspond to distinct group elements. This leads to conflicting gradients, resulting in the imperfectness of the latent encoding in ERL.

## J ADDITIONAL QUALITATIVE RESULTS

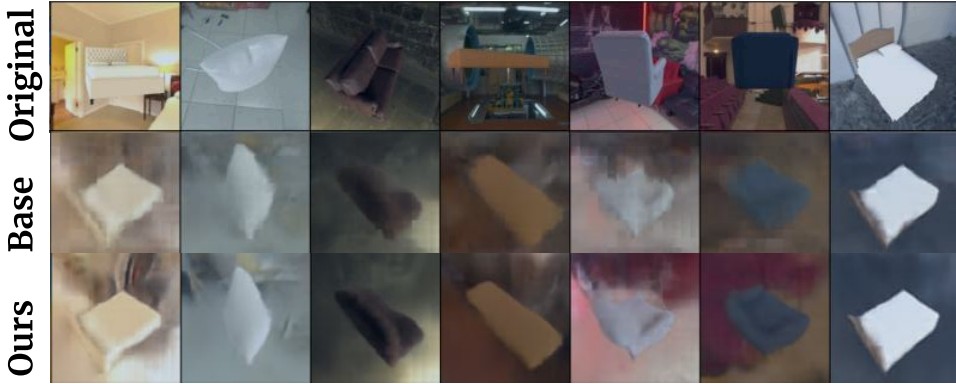

Figure III: Qualitative comparison of novel view synthesis on out-of-distribution (OOD) datasets with and without latent correction. Results from the ABO-Material OOD set.

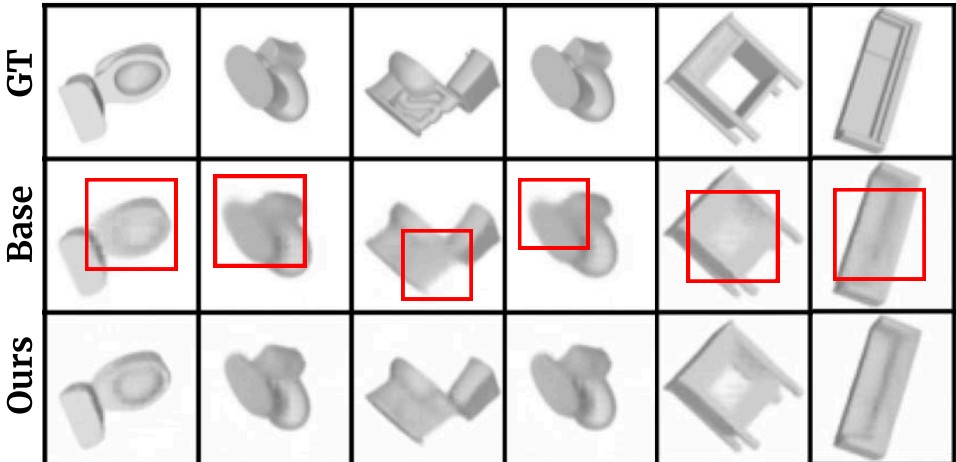

Figure IV: Qualitative comparison of novel view synthesis on out-of-distribution (OOD) datasets with and without latent correction. Results from the ModelNet10-SO(3) OOD set.

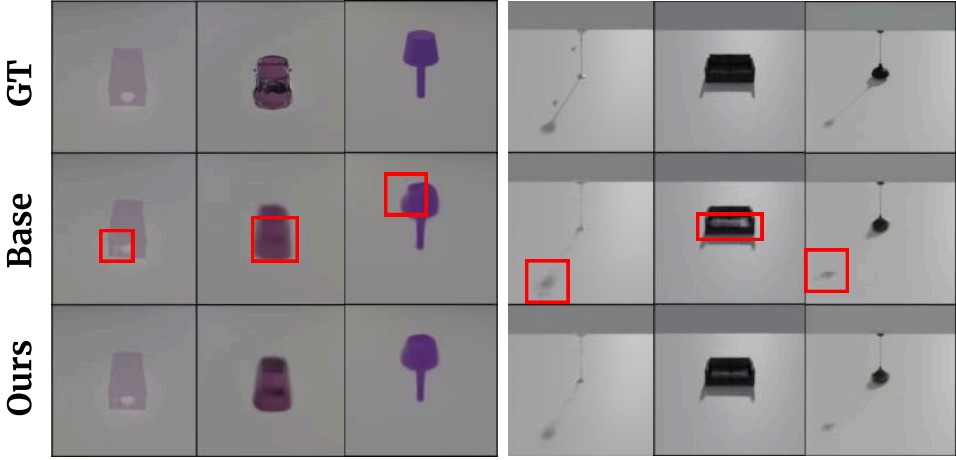

Figure V: Qualitative comparison of novel view synthesis on out-of-distribution (OOD) datasets with and without latent correction. **Left:** Results from the ComplexBRDFs OOD set. **Right:** Results from the ABO-Material Day-to-Night OOD set.

