# OpenReview forum: "Equivariant Latent Alignment via Flow Matching under Group Symmetries"
_ICLR.cc/2026/Conference — Submitted to ICLR 2026_

### Official Review · Reviewer_JC1E · 2025-10-30

**Soundness:** 3
**Presentation:** 3
**Contribution:** 3
**Rating:** 8
**Confidence:** 3

**Summary:**

This paper introduces Residual Latent Flow to correct latent misalignment in equivariant representation learning. The flow-matching approach enhances latent alignment and novel view synthesis while preserving group structure across SO(2) and SO(3) symmetries.

**Strengths:**

- combines flow matching with equivariant representation learning to address a previously underexplored problem.
- well-structured, with clear motivations, derivations, and visualizations.
- offers a practical and theoretically grounded solution to a fundamental limitation in ERL, with potential impact on generative modeling and view synthesis.
- extensive experiments across multiple datasets, metrics, and model sizes.

**Weaknesses:**

- the method assumes access to a pre-trained equivariant encoder and decoder, which may limit its applicability if such models are not available or poorly trained.
- the flow model adds computational overhead, though the paper shows that even a smaller model brings improvements.
- while the method improves alignment, the final synthesis quality is still bounded by the decoder’s capacity.

**Questions:**

- how does the method scale to higher-dimensional or non-compact groups beyond SO(2) and SO(3)?
- could the flow correction be integrated into the encoder training loop, rather than applied post-hoc?
- the paper uses a linear interpolation path. have you experimented with stochastic paths, and if so, how did they affect performance?
- the decoder is fine-tuned on flow-corrected latents. did you observe any overfitting or degradation when fine-tuning was omitted?

---

> ### Author Response · Authors · 2025-11-28
>
> We appreciate the reviewer for valuable feedback. Please see the detailed answers for each question below.
>
> __(W1,3) Reliance on pre-trained equivariant encoder and decoder__
>
> Yes, we agree that our method relies on a pre-trained ERL model, but since there are several powerful ERL frameworks (NFT, LGA, Dupont et al., Sajjadi et al.), this would not be a serious problem. On top of that, our paper aims to address the latent misalignment issue by carefully refining their latent representations with flow matching. We emphasize that these autoencoders are easy to train and expressive enough, allowing us competitive performance. Please refer to Tab. 2 for comparison with various ERL baselines on the RotatedMNIST reconstruction task. As the reviewer commented, the final synthesis quality would be further improved when a more capable decoder is provided, but this is an orthogonal direction for this task, not within the scope of this work.
>
> ---
>
> __(W2) Computational overhead__
>
> We emaphasize the extra effort for training this generative flow model for correction is relatively marginal, compared to the cost required to train the original autoencoder for equivariant representation learning. For example, in Tab. III and IV in Appendix G, we provide the number of parameters of the autoencoder (120M) + flow model (2M). Even with this small overhead (1.7%), our method achieves significant gain in PSNR (22.73 → 24.32) and prediction error (0.0056 → 0.0039). We added a few more extra datasets in the revised Tab. 1, demonstrating meaningful improvement (e.g., reduced prediction error by 15-30%) in NVS with this small extra overhead on various datasets.
>
> ---
>
> __(Q1) Scalability to higher-dimensional / non-compact groups beyond SO(3)__
>
> For dimensionality scaling, we provide a scaling experiment comparing performance across various latent channel dimension ($C$) and maximum degree of representation ($L$) in Tab. III and IV, respectively, in Appendix G. We agree that extending our work beyond SO(3) would be an interesting future direction, but we put the focus of this work within the practical novel view synthesis, where the most natural group symmetry arises from rotational group within the 3D space (SO(2) and SO(3)).
>
> ---
>
> __(Q2) Potential extension to end-to-end training with the encoder__
>
> We appreciate the reviewer for this insightful question. We in fact tried end-to-end training but the proposed post-hoc training turned out to be more effective and stable. We observe that training both the encoder and corrector at the same time is unstable, probably due to its significantly higher degree of freedom. That is, the end-to-end training keeps changing the prior and target distribution, so the flow matching model is unable to stably learn a transportation between them. For this reason, we conclude that freezing the pre-trained encoder and training only the flow model post-hoc is a more stable way in practice.
>
> ---
>
> __(Q3) Stochastic path__
>
> As suggested, we additionally conducted an ablation study with stochastic paths in Appendix F, but we still find out the deterministic path is more effective to learn our latent correcting flow model. Our interpretation is that the additional stochasticity does not significantly enrich the target distribution in our correction scenario, but increasing the noise level merely enlarges the variance of the training trajectories around the same endpoints, which acts as label noise for the velocity field. Because the flow model learns a genuinely object-specific vector field, injecting stochastic noise induces interference between probability paths corresponding to different objects, possibly causing performance degradation. If our method behaved like standard flow matching with arbitrary, non-structured transportation, such degradation would not appear, just as shown in prior work on stochastic interpolants. Please see Appendix F for more details.
>
> ---
>
> __(Q4) Overfitting or degradation without fine-tuning__
>
> Yes, we actually observed image quality degradation when we use un-finetuned decoder, motivating us to introduce the fine-tuning stage. Since the flow-corrected latents follow a shifted distribution compared to the analytically rotated ones used at pretraining, we allow the decoder to update its parameters to fairly adapt to this new latent space. This ensures both the autoencoder and flow models to operate under consistent decoder behavior. We elaborate this fine-tuning stage in the revised Sec.3.3, with our empirical discovery.

---

### Official Review · Reviewer_jAkP · 2025-10-31

**Soundness:** 3
**Presentation:** 3
**Contribution:** 3
**Rating:** 6
**Confidence:** 3

**Summary:**

This paper identifies latent misalignment in equivariant representation learning(ERL). It means a persistent gap between the analytic group action expected in the latent space and the encoder output for the transformed input. The gap grows with larger rotations and harms geometric consistency and novel view synthesis quality. The authors propose Residual Latent Flow, a conditional flow matching correction that uses the same object and the same group element to transport an analytic latent to its paired target latent along a zero variance linear path while learning a velocity field that points from the source to the target. Integrating this field at inference maps the analytic latent to the correct target and keeps the intended group structure, which is implemented with an NFT based block diagonal parameterization of irreducible components. On synthetic benchmarks with planar and three dimensional rotations, the method improves image reconstruction metrics and reduces a latent angle error derived from the first order component, producing more faithful and symmetry aware novel view synthesis.

**Strengths:**

The paper clearly defines latent misalignment in equivariant representation learning and ties it to drops in geometric consistency and synthesis fidelity, then motivates a solution by identifying the conflict between the equivariance loss and the decoder loss. Built on Neural Fourier Transform, it uses a block diagonal, interpretable decomposition into irreducible components for fine grained diagnosis. The proposed Residual Latent Flow is a simple yet principled correction that conditions source and target on the same object and transform, learns a residual transport in latent space, and preserves group structure. Experiments on SO(2) and SO(3) show consistent reductions in misalignment and gains in reconstruction and novel view synthesis, indicating the method complements rather than replaces ERL.

**Weaknesses:**

From the method as written, I point out several potential weaknesses.
1. The reliance on a linear, zero-noise interpolation path may break under large rotations or strong lighting and occlusion shifts; without comparisons to curved or noise-injected paths, I remain unsure about error accumulation and robustness.
2. The method assumes paired supervision with the same object and the same transform, yet real data often has label noise or imperfect pairings; without mitigation strategies, I worry the learned correction becomes biased and brittle.
3. The evaluation centers on a degree-one block angle error, which can miss higher-order misalignment and bake in encoder bias; without multi-block or encoder-agnostic metrics, I doubt the coverage of the evidence.
4. As latent dimensionality and group complexity grow, compute, memory, and optimization stability may bottleneck; without scaling curves and throughput or latency reports, I remain uncertain about practical viability.

**Questions:**

1. How sensitive are results to using a linear, zero-noise path? Do small positive noise or non-linear paths materially change alignment and synthesis at larger rotations or under lighting/occlusion shifts?
2. How robust is the method when same-object, same-transform pairs are imperfect (label noise, calibration drift, identity mixups)? What failure modes appear, and are there mitigation strategies?
3. Does the evaluation capture misalignment in higher-order components, not just the degree-1 block? Can you report multi-block or block-weighted summaries and encoder-agnostic checks?
4. Is the method still practical in high-dimensional latents and more complex groups? Please analyze the practical feasibility of applying this method to more complex structures.

---

> ### Author Response · Authors · 2025-11-28
>
> We appreciate the reviewer for valuable feedback. We hope our updated manuscript answers your questions.
>
> __(W1,Q1) Sensitivity to nonlinear / noisy paths__
>
> We would like to first clarify that our method primarily aims to correct the latent representation induced from a pre-trained autoencoder, which refines the latents to satisfy the equivariance relation more strongly. Hence, the flow model acts as a corrector or a refiner to the latents, rather than directly generating the full path corresponding to a NVS path. If the flow model were intended to infer the NVS trajectory itself, we agree that a curved path would be a more natural choice. However, because our objective is to use flow matching purely as a corrector, we posit that linear paths are potentially sufficient for the correcting process. Indeed, Fig. 4 implies that even under large rotational transformations, our linear-path-based corrector effectively refines the latent representation to significantly better satisfy the equivariance relation than the baseline, yielding consistently superior PSNR. We believe this empirical evidence supports our claim that a linear-path flow model serves as a robust corrector across the full range of rotation degrees.
>
> Regarding the comment on noisy path, we additionally performed an ablation study on stochastic paths to train flow matching in Appendix F. As seen in Tab. II, we observe that the deterministic path is the most effective way to learn our latent correcting flow model. This performance gap further supports that our object-aware conditional path behaves as intended. Because the model learns a genuinely object-specific vector field, injecting stochastic noise induces interference between probability paths corresponding to different objects, possibly leading to performance degradation. If our method behaved like standard flow matching with arbitrary, non-structured transportation, such degradation would not appear, just as shown in prior work on stochastic interpolants; random perturbations do not harm models that do not rely on semantically aligned trajectories.
>
> ---
>
> __(W2,Q2) Robustness on noisy environment__
>
> We appreciate the reviewer for this insightful question. We conducted an additional experiment on noisy label settings, and report the result in Tab. V in Appendix H. Specifically, we corrupted the labels of the 10% training samples by 2.5, 5, 7.5, and 10 degrees, and measured prediction accuracy of NVS. We observe that both our method and the NFT baseline perform reasonably well unless the noise level is relatively high (e.g., larger than 10$^\circ$), but our method consistently outperforms NFT under all noise levels. We agree that further reducing this performance degradation by a mitigation strategy would be an interesting future direction.
>
> ---
>
> __(W3,Q3) Additional metrics beyond degree-one block angle error__
>
> For the evaluation metrics to capture higher degree components as well in SO(2), we modified the definition of angle error in Eq. (17) in Sec. 4.2. We first estimate the angle that minimizes the Frobenius norm discrepancy for each ℓ, and take the average over every ℓ. This now incorporates misalignment in higher order components. For SO(3), the degeneracy of the Wahba problem's solution complicates to compute the angle error for higher degree vectors. Hence, we additionally compute latent error, which is the L2 distance between analytically rotated (+flow-corrected) latent and the ground-truth latent. This will be another metric that captures the misalignment across all higher degree representations. The revised Tab. 1 demonstrates the efficacy of our method across all metrics.
>
> ---
>
> __(W4,Q4) Scalability to higher-dimensional / more complex groups__
>
> In Tab. III and IV in Appendix G, we provide scaling experiment for increasing the latent channel dimension $C$ (latent dimensionality) and the maximum degree of representation $L$ (group complexity), reporting the latency (ms/sample) and the number of parameters for each model. We also provide Fig. II, demonstrating stable optimization curve for more complex models. We also report the training cost of the flow matching component including per-step runtime and memory usage in both SO(2) and SO(3) setups in Appendix G.

---

### Official Review · Reviewer_Q8pz · 2025-11-01

**Soundness:** 3
**Presentation:** 3
**Contribution:** 3
**Rating:** 6
**Confidence:** 1

**Summary:**

This paper addresses the problem of latent misalignment in equivariant representation learning for geometry-aware generative models and novel view synthesis (NVS).

The author address this problem by adding another flow matching process to correct the analytically rotated latents to their corresponding empirically encoded targets. Additionally, the author can finetune the decoder to take the flow-matching corrected latents to reconstruct target signal, which could further improve performance.

The author evaluate the methods a several synthetic image benchmarks, and shows that the proposed methods improves both alignment accuracy and novel view synthesis quality compared with baseline equivariant encoders.

To confess, I didn't fully understand what's the relationship between equivelanet representation and task like novel view synthesis, and how the methods compares to popular methods like NeRF and 3D GS. So, I would say I have low confidence in my review of the paper.

**Strengths:**

1. The paper address a very critial problem in equivelanet representation learning: the mismatch between analytically rotated latents to their corresponding empirically encoded targets.  And the author address the problem with a method that makes a lot of sense, train a generative model (flow matching) to correct it in latent space.
2. The results seems quite good, though its at pure syntheic dataset.

**Weaknesses:**

1. I find it slightly hard to understand what's the input, and how is latent analytically rotated, and what's the training loss, untill I read the supplementary.  Maybe moving some details to 4.1 or 3.2 from Appendix B.3 might make reader understand easier?
2. The author should include some baseline results on NVS, rether than only comparing with yourself, and the vanilla baseline.  Also, the PSNR in Figure-4 seems comfusing, it will be better to show the absolute number.

**Questions:**

1. So, do I understand it correctly, by training on a dataset of tens of objects, each with multiview inputs. What you are trying to do is kind of letting the model maybe mermoize the objects multiview, and also get a latent space that enables you to traverse over the viewpoint sphere?
2. If my understanding is kind of correct, the equivelanet representation should not be able to generate nove views of unseen objects during training.  Cause generate nove views of unseen objects during training is a probabilistic task, and seems the representation space and the decoder is determinstic model, which does not has the ability to sample novel views, especially unseen part of the object?
3. I kind of don't understand why this equivelanet representation would help for novel view synthesis?  and how does it compare with NeRF and 3DGS?

---

> ### Author Response · Authors · 2025-11-28
>
> We appreciate the reviewer for pointing these out and providing valuable feedbacks. We hope our answers satisfactorily resolve your concerns.
>
> __(W1) Moving some details from Appendix B.3__
>
> We appreciate the reviewer for this constructive feedback. We updated our manuscript to bring up the training details to the method section (Sec. 3.3) and refined the descriptions to more clearly present our loss function and training method. We also expanded the Sec. 3.1 for more explanations of NFT, to elaborate how the group elements are represented and latents are analytically rotated.
>
> ---
>
> __(W2) Baseline comparison on NVS__
>
> Following the reviewer's suggestion, we conducted additional experiments to compare with various baseline methods for NVS such as Spatial-VAE, GIAE, LGA, and NFT on RotatedMNIST. As seen in Tab. 2, our proposed method consistently outperforms all baselines across various metrics. We also added additional datasets consisting of real images in Tab. 1 for more comprehensive experimental comparison. We also changed the Fig. 4 to depict the absolute value of PSNR, reflecting the reviewer's comment.
>
> ---
>
> __(Q1,2) Novel view synthesis by deterministic decoder during training__
>
> We would like to respectfully clarify that a deterministic model is still fully capable of synthesizing novel views of an unseen object, provided that the encoder and decoder are able to learn the correct equivariance and inductive biases. While we agree with your intuition about memorization, the introduced equivariant representation learning is not intended to only memorize multiview observations, but it also learns a structured latent space that generalizes even for unseen test objects and view angles.
>
> To verify this, we conducted additional experiments on two test data splits: (i) out-of-distribution (OOD) object, where the object itself has never been shown during training, and (ii) unseen angle for in-distribution object, where the object itself has been shown during training but the queried angle at test time is unseen. Our method successfully predicts novel view for both cases, as described in the revised Sec. 4.3.2. This result indicates that with sufficient geometric consistency in the representation, the model reliably predicts unseen views without requiring any probabilistic sampling.
>
> ---
>
> __(Q3) Why does equivariant representation help novel view synthesis?__
>
> Equivariant scene representations impose geometric structure, such as consistent behavior under rotations or viewpoint changes, leading to lower-variance gradients and more stable optimization. By encoding geometry in a group symmetry-aware way, they reduce ambiguity during training and thus potentially generalize better to unseen views. This places them between the (vanilla) NeRF’s weakly constrained implicit field with almost zero geometry priors and 3DGS’s fully explicit primitives (a set of Gaussians explicitly defining the geometric structure of the object), offering a balance of robustness, inductive bias, and high-quality novel-view synthesis.

---

### Official Review · Reviewer_PQ7u · 2025-11-01

**Soundness:** 3
**Presentation:** 4
**Contribution:** 2
**Rating:** 4
**Confidence:** 4

**Summary:**

- In latent-to-image equivariant generative models of images, it is expected that an analytical transformation of the latent will lead to the exact corresponding transformation of the generated image. E.g., rotating the latent should produce the corresponding rotation in the output image.
- However, this never occurs exactly in practice and the paper posits that this is because the latent must encode non-pose related information such as lighting.
- The paper proposes a framework that maps the latents to new latents that produce exactly the desired transformation when decoded. It does so by using a flow matching generative model.
- Experiments are presented against a baseline on three representative object-centered synthetic datasets with exact control over the appearance and pose of the objects. They show that the proposed latent realignment approach produces higher-quality object transformations to unobserved poses.

**Strengths:**

- The paper is quite well written and explained. It is also refreshingly does not read like it was over-edited by LLMs, making it much easier and more enjoyable to read.
- Extreme effort has been put into making the paper as accessible as possible to new readers and it is appreciated.
- The figures are very strong as they get the main thesis of the paper across well.
- Designing a posthoc correction algorithm for latents in an equivariant model to better respect equivariance constraints is an interesting idea in general.

**Weaknesses:**

### Methodology and framing:
The paper's framing is confusing. It (A) considers a simple autoencoder with an equivariance loss to be the end-all in equivariant representation learning and (B) assumes that the quality of representations should be evaluated solely by considering whether the decoded image was correctly transformed. While I'm sure that this approach existed, it is not cited anywhere in the main text (as far as I can see) and it also appears to be the sole baseline.
- Regarding (A): There are many equivariant representation learners for image synthesis that do not take this form and it is unclear if the proposed methodology would extend to them.
- Regarding (B): Network representations are commonly evaluated using a diverse set of downstream tasks and only image synthesis (on synthetic datasets) is presented here. For example, what about equivariant representations for other tasks that may benefit such as pose estimation?  Further, the focus on image synthesis raises non relevant but practically material concerns like the experimental trends very much depending on decoder capacity, latent size, the image loss used, etc.

Currently, the gains displayed in Table 1 are marginal relative to the effort required to train a secondary posthoc generative flow model to improve the original model. It is unclear if these small gains will transfer to more powerful representation learning or generative modeling strategies.

Lastly, as a more minor point, L053 claims that some portion of the empirical lack of equivariance is due to the data variability in non-closed-form ways (e.g. occlusion, lighting). While this is true to some extent, even with a perfect dataset, this empirical lack of equivariance will always persist with current architectures due to aliasing of intermediate representations (see [1](https://arxiv.org/abs/2106.12423) and [2](https://arxiv.org/abs/2210.02984)). This statement in the paper should be given more nuance.

### Experiments:
- All of the experiments in the paper are conducted on entirely synthetic programmatically generated images. While these are great for benchmarking as they give exact ground truth, IMO they are insufficient to demonstrate that a given algorithm is useful for real problems. This paper would need to design and execute experiments with a real dataset or two before it can be considered to be broadly useful.
- As far as I can tell, there are no baselines in the entire paper, just a single uncited method that just removes the flow matching from the proposed method (i.e. an ablation). This is insufficient and the paper needs significantly more experimental depth. There are many, many papers in this space about equivariant latents creating controllable outputs such as [1](https://proceedings.neurips.cc/paper_files/paper/2024/file/e63309e532688c722177f81e99f94f32-Paper-Conference.pdf), [2](https://proceedings.neurips.cc/paper_files/paper/2023/file/9bfc2c20fa2f56a18397eafe1be8a50a-Paper-Conference.pdf), [3](https://proceedings.mlr.press/v162/birrell22a.html), [4](https://arxiv.org/pdf/1909.11663), [5](https://arxiv.org/pdf/2202.07559), [6](https://arxiv.org/pdf/2008.11673), [7](https://arxiv.org/abs/2106.12423), and many others including all the references within them. At least two to three relevant baselines should be included.

### Presentation:
- From Section 3.2, it is clear *how* the flow matching is performed. However, unless I missed something, the paper never clarifies *why* it decided on a flow matching setup. One can imagine many different ways to achieve this latent alignment, and flow matching is not the first approach that would come to mind. Please elaborate on the rationale for this choice and its comparative advantage over other approaches for this problem.
- It is quite unclear from the paper whether the flow matching module is trained alongside the image representation learner end-to-end or if its trained post hoc. Please clarify.
- The paper takes too long to get to its contribution (on page 5) as it covers the reasonably well known fundamentals of equivariant learning in great detail. As a result, many key details that are much more specific to the paper, like Appendices B.2--B.4 are missing from the main text. The background of their chosen model based on this relatively new strategy called "Neural Fourier Transforms" should be expanded as well.

**Questions:**

It is possible that I misunderstood core contributions and I am very open to being corrected during the discussion period. As of now, the experiments come off as too preliminary for this venue and the methodological setup is confusing (see above for specifics) -- please focus on these aspects in the rebuttal.

---

> ### Author Response · Authors · 2025-11-28
>
> We sincerely appreciate the reviewer's constructive feedback, mainly regarding the problem formulation and lack of baseline comparison. These comments indeed greatly helped us to improve our paper. We provided detailed answers and additional experimental results suggested by the reivewer below.
>
> > __Methodology and Framing__
>
> __1. Clarification on problem formulation__
>
> First of all, we appreciate the feedback for unclear problem setting, and in response to this, we clarified the novel view synthesis (NVS) task we are tackling in Appendix A. Specifically, we are interested in the most-general NVS setting under SO(3) symmetry, where we aim to reconstruct an object seen from another viewpoint in any direction within the 3D space; or equivalently, an object can rotate both to the longitudinal and latitudinal directions given a fixed camera direction. This general setting encompasses special cases where the object can rotate only through one direction, e.g., in-plane rotation where the rotating axis is parallel to the camera direction, and out-of-plane rotation (but restricted to SO(2) symmetry; i.e., camera view can rotate only with respect to single axis) where the rotating axis is not parallel to the camera direction (typically the two axes are perpendicular to each other in SO(2)). These problem settings are illustrated in Fig. I in Appendix A.
>
> __(A) Applicability to other ERL frameworks & Why autoencoders for equivariant representation learning?__ We agree that equivariant representation learning spans various methods beyond autoencoders. Our main idea is to use flow matching as a corrector to the latent representations, and we believe our method is generally applicable to any latent space-based NVS model which uses differentiable encoder and decoder, not just autoencoders. The main reason for us to focus on autoencoders is that we are tackling the general NVS task under SO(3) symmetry described above. Unlike its simpler special case like in-plane rotation, autoencoders have been dominantly adopted to learn equivariant latent spaces, primarily because the model should compress visual information into a latent code that supports reasoning about unobserved or occluded viewpoints, and autoencoders are natural and suitable for this purpose. For example, NFT [A], LGA [B], and ENR [C] first encode an image into a latent representation, apply the group action, and decode it to back to another image. As a response to the reviewer's suggestion, we additionally compare in Tab. 2 with these methods on smallNORB (out-of-plane-rotation synthesis, SO(3)). We also add another experiment in Tab. 3 comparing with the previous NVS methods which have not used autoencoders on RotatedMNIST (in-plane rotation synthesis).
>
> __(B) Downstream tasks beyond image synthesis.__ While we focused on verifying the hypothesis ""learning most equivariant representation is useful under the NVS task"" in this work, we agree that it would be interesting to see whether the learned equivariant representations could be also useful in other downstream tasks such as pose estimation.
>
> To address this point, we include an additional experiment in Appendix E, examining whether a lightweight regression head (a 2-layer MLP) would be able to extract rotation information from the learned latent representations by predicting (i) the original rotation angle of the input image and (ii) the angle of internal rotation acted in the latent space. As seen in Tab. II in Appendix E, our method yields a clear improvement over NFT, indicating that the learned representations are potentially useful beyond image synthesis. We view this experiment as a first step, and extending such downstream evaluations to a wider set of tasks would be a valuable future direction.
>
> ---
>
> __2. Gains relative to the effort to train post-hoc flow model__
>
> We emaphasize the extra effort to train the flow model is actually marginal, compared to the cost required to train the original autoencoder for equivariant representation learning. For example, in Tab. III and IV in Appendix G, we provide the number of parameters of the autoencoder (120M) + flow model (2M). Even with small overhead (1.7%), our approach achieves significant gain in PSNR (22.73 → 24.32) and prediction error (0.0056 → 0.0039). We added a few more extra datasets in the revised Tab. 1, demonstrating meaningful improvement (e.g., reduced prediction error by 15-30%) in NVS with this small extra overhead on various datasets.
>
> ---
>
> __3. Empirical lack of equivariance__
>
> We thank the reviewer for pointing out this valuable point. We indeed agree with this point, and updated our Introduction (line 54-58) accordingly to provide the context about aliasing problem due to architectures and the related literatures.

---

> > ### Author Response · Authors · 2025-11-28
> >
> > > __Experiments__
> >
> > __1. Experiment on real dataset__
> >
> > To verify applicability of our method on real datasets, we conducted additional experiments on RotatedMNIST and smallNORB. The smallNORB contains images of 50 toys belonging to 5 generic categories, captured by two cameras under 6 lighting conditions, 9 elevations (30 to 70 degrees every 5 degrees), and 18 azimuths (0 to 340 every 20 degrees). We report the evaluation results for these real-world datasets in Tab. 1 in Sec. 4.3.2, indicating that our proposed method meaningfully improves the performance across all metrics. We hope this experiment verifies the usefulness of our method not just on programmatically generated data, but on the real problems.
> >
> > ---
> >
> > __2. Comparison with baselines__
> >
> > We first sincerely thank the reviewer for this constructive feedback and for providing a valuable list of potential baselines. It does help a lot to further verify the effectiveness of our method. We conduct the suggested experiments comparing a subset of these methods, or provide an argument if we believe it is not a proper baseline in the NVS task.
> >
> > - __Fully comparable.__ LGA [1]: Like NFT, this work learns an autoencoder that learns equivariance relation. This method decomposes the latent vectors into group-invariant and group-equivariant components, and then manipulates the equivariant part only. This method is indeed a comparable baseline to our method, and thus following the reviewer's suggestion, we compare with this method on in-plane rotation synthesis and out-of-plane rotation (SO(3)) in Tab. 3. Additionally, we found ENR [C] as another valid baseline to compare with, so we add this on the out-of-plane rotation (SO(3)) synthesis experiment in Tab. 3. Orientation-disentangled unsupervised learning [6] decomposes latent vectors into group-invariant and group-equivariant components, similarly as in LGA [1]. However, this work relies on a particular SE(2)-equivariant CNN architecture within a VAE. We believe a substantial modification would be required to make it applicable to the general NVS setting, but we believe LGA [1] would be a more general and stronger baseline sharing a similar core idea. Due to the limited rebuttal period, we focus on the stronger baselines instead of including it.
> >
> > - __Partially comparable.__ Spatial-VAE [4] predicts the pixel intensity at each corresponding coordinate from spatial coordinates and latent variables, following a concept similar to implicit neural representations. This method explictily acts SO(2) rotation matrix upon the coordinate system and then predict the pixel intensity on that rotated coordinate system. Group-invariant AE [5], uses an autoencoder to predict a canonical image, which serves as a representative of the orbit it belongs to, and then manually rotates the canonical image to predict a rotated image. These methods are applicable to the in-plane rotation, so we compare with them on RotatedMNIST in Tab. 2. However, they do not extend to out-of-plane rotations like SO(3), as their input is constrained to the given 2D coordinate system.
> >
> > - __Not comparable.__ FFRL [2] is about learning flow factorized VAE, primarily aiming at latent factor disentanglement, and structure-preserving GAN [3] proposes a data-efficient framework for learning distributions with additional structure such as group symmetry. Following unsupervised approaches, however, these two methods do NOT provide controllable manipulation (e.g., rotation with an explicit angle). Therefore, although they aim to learn an equivariance-aware latent space, we believe they are not directly applicable to and comparable on the NVS task. StyleGAN3 [7] also designs an equivariant network, but this work is mainly about training GANs, instead of tackling the NVS task with controllable generation. Thus, we believe this is also not directly comparable on the given NVS task.
> >
> > Overall, most controllable generation methods, except for LGA, focus on solving a disentangling problem such as rotating MNIST images or biological images under SO(2) or SE(2) symmetry group. These are in-plane rotation synthesis tasks, which are considerably simpler than our NVS task setting, predicting an image from an unseen viewpoint in 3D (e.g., backview of an unseen chair). We respectfully argue that only LGA [1] and ENR [C] are fully comparable for this task, but we are open for further discussion. To this end, we provide additional experiment on RotatedMNIST for SO(2) in-plane rotation synthesis, comparing with LGA, Spatial-VAE, GIAE, and NFT in Tab. 2. For out-of-plane rotation synthesis, we compare LGA, ENR, NFT, and ours on out-of-plane rotation synthesis (SO(3)) in Tab. 3.

---

> ### Author Response · Authors · 2025-11-28
>
> > __Presentation__
>
> __1. Why flow matching?__
>
> We view the task of correcting equivariant latent representations as a distribution transport problem, because we are trying to match $\rho(gh^{-1}) \Phi(h \circ x_0)$ with $\Phi(g \circ x_0)$ for every $g, h \in G$, i.e., there exists multiple source points $\{ \rho(gh^{-1}) \Phi(h \circ x_0) \}_{h \in G}$ which needs to match with a single target point, for every $g \in G$. To clarify this motivation, we added Fig. 2(c) in the revised manuscript. As seen in this illustration, our model aims to transport the distribution of purple and yellow source samples to the target distribution of blue points.
>
> This setup is inherently a transport problem, and we found the flow matching framework particularly compelling due to its principled and effective mechanism for learning such mappings. Unlike direct regression, which might seem like a natural first attempt, flow matching provides an iterative mapping that can flexibly adapt to arbitrary source and target distributions. A diffusion-based approach may come to mind as an alternative for iterative refinement, but it is restricted to starting from a Gaussian prior. Since this task requires transporting latents from an arbitrary distribution, it is not a fully compatible approach for this task. With the flexibility of prior distribution, we conclude that flow matching is an compelling approach to tackle this probelm.
>
> We hope the revised Fig. 3 helps convey our reasoning and demonstrates why the flow matching approach is well suited to the task.
>
> ---
>
> __2. End-to-end vs. Post-hot training__
>
> Our model is trained post-hoc. We first train the autoencoder with the ERL loss in Eq. (1), and then the flow model with the frozen encoder for latent correction. We added Sec. 3.3 to state more clerly about this training process. We tried end-to-end training but the proposed post-hoc training turned out to be more effective and stable. Training both the encoder and corrector at the same time is unstable, since the end-to-end training keeps changing the prior and target distribution, and thus the flow matching model is unable to stably learn a transportation between them.
>
> ---
>
> __3. Presentation for key contributions__
>
> Following the reviewer's suggestion, we updated our manuscript to bring up the training details to the Method section (Sec. 3.3) and refined the descriptions to more clearly present our training method. We also expanded Sec. 3.1 for more explanations of NFT, as this serves as our baseline model.
>
> ---
>
> We thank again the reviewer for the detailed comments and critical feedback. Please feel free to reply if there is any uncovered issue or things to discuss. We will reply to additional questions in timely manner.
>
> [A] Koyama, Masanori, et al. "Neural fourier transform: A general approach to equivariant representation learning." ICLR (2024)
>
> [B] Jin, Yinzhu, Aman Shrivastava, and P. Thomas Fletcher. "Learning group actions on latent representations." NeurIPS (2024)
>
> [C] Dupont, Emilien, et al. "Equivariant neural rendering." ICML (2020)

---

### Author Response · Authors · 2025-12-02
**Final comment to AC and reviewers**

We sincerely appreciate AC for handling our submission under this unusual circumstance. Below, we summarize our answers to the questions and revision of the manuscript reflecting suggestions by the reviewers. Please see the full corresponding rebuttals for more details.

---

__1. Comparison with baselines are insufficient. (PQ7u, Q8pz).__

Following the reviewers' suggestion, we add comparisons with NFT, LGA, ENR, Spatial-VAE, GIAE in Tab. 2-3. A few other suggested models are excluded since their problem formuation is not comparable. For this, we provide a clearer distinction between in-plane, out-of-plane, and full SO(3) rotations in Appendix A.

__2. Experiments on real dataset with noisy labels are recommended. (PQ7u, jAkP)__

We add experiments on RotatedMNIST and smallNORB composed of real objects (Tab. 1, Tab. 2, Tab. 3, Fig. 5), demonstrating consistent improvements with our method across all metrics as suggested by PQ7u. We also empirically verify robustness of our method under label noise (Appendix E.3) as suggested by jAkP.

__3. Provide justification of using flow matching, instead of simpler alternatives. (PQ7u)__

We motivate the latent alignment as a distribution-transport problem and add a clarifying illustration (Fig. 2c). We clarify why flow matching is an appropriate approach to solve this probelm, particularly regarding its flexibility of prior distribution.

__4. Additional computational cost to train the post-hoc flow model is unclear. (JC1E, jAkP)__

Our flow-based correction yields strong gains (e.g., PSNR 22.73 → 24.32, error 0.0056 → 0.0039) by adding only 1.7% parameters on top of the base model. Following jAkP's suggestion, we report scaling experiments, showing smooth and stable performance improvement with larger latent dimensions and increased group complexity. (Appendix G, Tab. III–IV)

__5. Ablation on stochastic paths is recommended. (jAkP, JC1E)__

Following the reviewers' suggestion, we include ablation study on stochastic vs. deterministic paths in Appendix F, Tab. II, confirming that our deterministic flow paths consistently outperform stochastic variants. We argue this is thanks to the elimination of interference between the object-conditional probablity paths when using the deterministic paths.

__6. How would the learned representations perform on downstream tasks beyond image synthesis? (PQ7u)__

We additionally evaluate on pose (rotation) prediction task using a light regression head (Appendix E, Tab. II) as, verifying clear gains over NFT.

__7. Misc. comments to improve presentation (PQ7u)__

We reorganize and clarify the method, expanding the description of NFT (Sec. 3.1), and move the training details (post-hoc training, fine-tuning stage) into Sec. 3.3, as per the request of PQ7u. We also add discussions of aliasing and architectural limitations as an additional cause of empirical lack of equivariance in introduction (line 055-058)."

---

### Meta-Review · Area_Chair_CAsL · 2026-01-05

**Summary:**

This paper proposes a flow-matching–based latent alignment approach to mitigate equivariance errors in generative models under known group transformations. The method treats analytically transformed latents as imperfect approximations and learns a residual flow to align them with empirically encoded representations. The approach is evaluated primarily on rotation-based novel view synthesis tasks, showing consistent but modest improvements over existing equivariant baselines.

While reviewers found the idea technically reasonable and the experiments carefully executed within the chosen setting, the overall consensus is that the scope of the contribution is narrow and the method does not constitute a sufficiently systematic or general advance. Concerns regarding limited applicability, incremental methodology, and unclear presentation ultimately informed the rejection decision.

**Reviewer Concerns:**

Concerns partially addressed by the rebuttal:
The rebuttal clarifies the motivation for latent misalignment and explains why analytically defined group actions may fail in practice due to encoder imperfections. Additional explanations of the flow-matching objective and training pipeline help clarify the mechanics of the proposed alignment procedure.

Existing concerns:
A primary concern across reviews is the limited scope of the work. The method is evaluated almost exclusively in controlled settings with explicit and known group transformations (e.g., rotations), and it remains unclear how the approach would extend to more realistic scenarios where symmetries are unknown, approximate, or data-dependent. As a result, the contribution appears specialized rather than broadly applicable.

Relatedly, reviewers noted that the proposed approach is not a systematic treatment of equivariant representation learning, but instead resembles a regularization or post-hoc correction mechanism. Learning a residual flow to align analytically transformed latents with empirical encodings is conceptually close to consistency regularization and latent refinement techniques, and does not introduce a new modeling framework or principle for enforcing equivariance.

In addition, several reviewers raised concerns about the clarity of presentation. Important implementation details of the model, training stages, and experimental settings are insufficiently specified, making it difficult to fully assess reproducibility and isolate the source of the reported gains. The experimental protocol and architectural choices would benefit from clearer and more structured exposition.

Finally, while empirical improvements are observed, the gains are relatively small compared to strong baselines, and it is unclear whether they justify the added model complexity and training overhead. These issues collectively limit the paper’s impact at the conference level.

**Reviewer Scores:**

Reviewer PQ7u: Likely to maintain their original score, as concerns about limited scope and incremental contribution remain unresolved.

Reviewer Q8pz: Likely to maintain their original score, given that the method is interpreted as a regularization technique rather than a systematic equivariant modeling approach.

Reviewer jAkP: Likely to maintain their score, viewing the work as technically correct but narrowly scoped and insufficiently general.

Reviewer JC1E: Likely to maintain their original score, citing unclear implementation details and limited empirical justification for the added complexity.

---

### Decision · Program_Chairs · 2026-01-26

Reject